# How Sparse Can We Prune A Deep Network: A Geometric Viewpoint

## Abstract

Network pruning constitutes an effective measure to alleviate the storage and computational burden of deep neural networks which arises from its overparameterization. A fundamental question is: How sparse can we prune a deep network without sacrifice on the performance? To address this problem, in this work we take a first principles approach, specifically, by directly enforcing the sparsity constraint on the original loss function and exploiting the universal *concentration* effect in the high-dimensional world, we're able to characterize the sharp phase transition point of pruning ratio, which equal one minus the normalized squared Gaussian width of a convex set determined by the Hessian matrix of the loss function. Meanwhile, we provide efficient countermeasures to address the challenges in computing the involved Gaussian width, including the spectrum estimation of a large-scale Hessian matrix and dealing with the non-definite positiveness of a Hessian matrix. Moreover, through the lens of the pruning ratio threshold, we're able to identify the key factors that impact the pruning performance. In specific, the flatter the loss landscape or the smaller the weight magnitude, the smaller pruning ratio. This result can provide unified and intuitive explanations on many phenomena of existing pruning algorithms. Extensive experiments are performed which demonstrate that the theoretical pruning ratio threshold coincides very well with the experimental one. All codes are available at: `https://anonymous.4open.science/r/Global-One-shot-Pruning-BC7B/`

## 1 Introduction

Deep neural networks (DNNs) have achieved stunning success in the past decade. The success of DNN relies heavily on overparametrization, i.e., the number of parameters are normally several order of magnitudes more than the number of data samples. Though being a key enabler for the striking performance of DNN, overparametrization however poses huge burden for computation and storage in practice. It is therefore much tempting to ask: Whether can we compress the DNN by a large ratio without no sacrifice of performance? and what's the limit of such model compressing?

To answer the first question, the main approach is to perform network pruning, which was first introduced by LeCun et al. (1989). Network pruning can substantially decrease the parameter number and thus alleviate the computational burden of inference and storage burden. The basic idea of network pruning is to devise metrics to evaluate the significance of parameters and then remove the insignificant ones. Various pruning algorithms have been proposed so far: LeCun et al. (1989); Han et al. (2015a;b); Luo et al. (2018); Zhou et al. (2016); Wang et al. (2018); Xiang et al. (2021); Molchanov et al. (2016); Li et al. (2016); He et al. (2019) and Han et al. (2015b).

In contrast, regarding the second question mentioned above, namely, the theoretical understanding of network pruning is unfortunately far less. Some relevant works are: Yang et al. (2023) explored the impact of network pruning on model's generalization ability. He et al. (2022) identified the relationship between the double descent phenomenon of network pruning and the learning distance. Larsen et al. (2021) proposed to characterize the degrees of freedom of a DNN by exploiting the framework of high-dimensional convex geometry.

Despite the above progress, it however still remains elusive about the *fundamental limit* of network pruning while maintaining the performance. To tackle this problem, we'll take a first principle approach by imposing the sparsity constraint directly on the loss function, thus converting the original pruning limit problem to a set intersection problem, i.e., deciding whether the *$k$-sparse set* intersects with the *loss sublevel set* (i.e., the set of weights whose corresponding loss is no larger than the original loss plus tolerance $\epsilon$) and obtaining the smallest value of $k$.

Intuitively speaking, the larger the loss sublevel set (higher complexity), the smaller the sparse set required for intersection, i.e., the network can be more sparse. To rigorously characterize the complexity of a set, by exploiting the high dimensional nature of DNNs, we are able to harness the notion of *statistical dimension* and *Gaussian width* in high dimensional convex geometry. Through this geometric perspective, it's possible for us to take advantage of the universal *concentration effect* (Vershynin, 2014; 2020) [1] in the high-dimensional world, so as to get sharp results about the above set intersection problem. In specific, we will exploit the powerful Approximate Kinematics Formula in high-dimensional geometry, which roughly says that for two convex cones, if the sum of their statistical dimension exceeds the ambient dimension, then these two cones would intersect with probability 1, otherwise they would intersect with probability 0. We notice that a sharp phase transition emerge here, thus enabling a precise and succinct characterization of the fundamental limit of network pruning.

The key **contributions** of this paper can be summarized as follows:

1. Our work is the first to characterize the fundamental limit of network pruning, whose result turns out to be both *precise* and *succinct*. The key message of this fundamental limit are twofold: 1) The smaller the network *flatness* (defined as the trace of the Hessian matrix), the more we can prune the network; 2) The smaller the *weight magnitude*, the more we can prune the network.

2. We provide an improved spectrum estimation algorithm for large-scale Hessian matrices when computing the Gaussian width of a high-dim. non-convex set.

3. We present intuitive explanations on many phenomena accompanied with existing pruning algorithms through our lens of the pruning ratio threshold, which include: (a). Why magnitude pruning is better than random pruning. (b). Why IMP requires iterative pruning instead of one-shot pruning. (c). Why gradually changing the pruning ratio during iterative pruning is preferred. (d). Why there exists significant performance difference in Rare Gems algorithm (Sreenivasan et al., 2022) between using and not using $l_2$ regularization.

## 1.1 RELATED WORK

**Pruning Methods:** Unstructured pruning involves removing unimportant weights without adhering to specific geometric shapes or constraints. Han et al. (2015b) presented the train-prune-retrain method, which reduces the storage and computation of neural networks by learning only the significant connections. Yang et al. (2017) employed the energy consumption of each layer to determine the pruning order and developed latency tables that employed greed to identify the layers that should be pruned. Guo et al. (2016) proposed dynamic network surgery, which reduced network complexity significantly by pruning connections in real time. Frankle & Carbin (2018) proposed pruning by iteratively removing part of the small weights, and based on Frankle's iterative pruning, Sreenivasan et al. (2022) introduced $l_2$-norm to constrain the magnitude of unimportant parameters during iterative training. Most of the existing pruning methods rely on heuristic algorithms, to delve into the fundamental constraints that impact network pruning, we adopt a first-principles approach by directly imposing sparsity constraints on the network and optimize for the convex $l_1$ constraint, which closely aligns with the sparsity constraint, the weights are pruned below a certain threshold. This is the most fundamental and straightforward pruning approach, providing valuable insights into the fundamental limit of network pruning research.

**Theoretical Advances in Understanding Neural Networks:** Despite a promising performance in empirical data, providing theoretical guarantees for neural networks remains challenging. Shwartz-Ziv & Tishby (2017) have explained the training dynamics of neural networks from the information theoretic perspective. The most prominent approach to understanding neural networks is the linearization or neural tangent kernel (NTK) technique Jacot et al. (2018). Using this linearization technique, it is possible to prove convergence to a zero training loss point. Additionally, Larsen et al. (2021) studied the training dimension threshold of the network from a geometric point of view, which shows that the network can be trained successfully with less degrees of freedom (DoF) in affine subspace, but the burn-in affine subspace needs a good starting point and also the lottery subspace is greatly affected by the principal components of the entire training trajectory. Therefore,

---

[1] Basically, the concentration effect says that a function of *a large amount* of independent (or weakly dependent) variables tends to concentrate to its expectation value. Notable examples include the Johnson-Lindenstrauss lemma (Bandeira et al., 2020) and related results in Compressive Sensing.

essentially the DoF result in Larsen et al. (2021) provides limited knowledge about the pruning ratio threshold, which is exactly the main subject of our work.

## 2 PROBLEM SETUP & KEY TOOLS

To explore the fundamental limit of network pruning, we'll take the first-principles approach as follows: by directly imposing the sparsity constraint on the original loss function and then pruning the trained weight vector by the magnitudes, the feasibility of pruning can thus be reduced to determine whether a sublevel set defined by the Hessian matrix of the loss function intersects a $k$-sparse set or a subspace. Through this framework, we're able to leverage powerful notions and tools in high-dimensional convex geometry, such as statistical dimension (Amelunxen et al., 2014), Gaussian width (Vershynin, 2014)and Approximate Kinematics Formula (Amelunxen et al., 2014).

**Model Setup.** Let $\hat{y} = f(\mathbf{w}, \mathbf{x})$ be a deep neural network $M$ with weights $\mathbf{w} \in \mathbb{R}^D$ and inputs $\mathbf{x} \in \mathbb{R}^K$. For a given training data set $\{\mathbf{x}_n, \mathbf{y}_n\}_{n=1}^N$ and loss function $\ell$, the empirical loss landscape is defined as $\mathcal{L}(\mathbf{w}) = \frac{1}{N} \sum_{n=1}^N \ell(f(\mathbf{w}, \mathbf{x}_n), \mathbf{y}_n)$. We employ classification as our primary task, where $\mathbf{y} \in \{0, 1\}^k$ with $k$ is the number of classes, and $\ell(f(\mathbf{w}, \mathbf{x}_n), \mathbf{y}_n)$ is the cross-entropy loss.

**Pruning Objective.** In essence, network pruning can be formulated as the following optimization problem:

$$\min \|\mathbf{w}\|_0 \quad \text{s.t.} \quad \mathcal{L}(\mathbf{w}) \leq \mathcal{L}(\mathbf{w}^*) + \epsilon \tag{1}$$

where $\mathbf{w}$ is the optimized weight and $\mathbf{w}^*$ is the original one. By utilizing the Lagrange formulation and convex relaxation of $l_0$ norm, Eq.1 can be reformulated as:

$$\min \mathcal{L}(\mathbf{w}) + \lambda \|\mathbf{w}\|_1 \tag{2}$$

After training with the above objective, the network weights will be pruned based on magnitudes, and the performance of the resulting network is evaluated. 5.

**Sparse Network.** The weight of the dense network is represented as $\mathbf{w}^*$, the weight of the sparse network, which retains the $k$ largest magnitude weights from $\mathbf{w}^*$, is denoted as $\mathbf{w}^k$.

**Loss Sublevel Sets.** A loss sublevel set of a network is the set of all weights $\mathbf{w}$ that achieve the loss up to $\mathcal{L}(\mathbf{w}^*) + \epsilon$:

$$S(\epsilon) := \{\mathbf{w} \in \mathbb{R}^D : \mathcal{L}(\mathbf{w}) \leq \mathcal{L}(\mathbf{w}^*) + \epsilon\}. \tag{3}$$

**Feasible $k$-Sparse Pruning.** We call $\mathbf{w}^k$ as a feasible $k$-sparse pruning if it obeys:

$$S(\epsilon) \cap \{\mathbf{w}^k\} \neq \emptyset, \tag{4}$$

and **the pruning ratio** is defined as $p = k/D$.

Below are some key notions and results from high dimensional convex geometry, which are of critical importance to our work.

**Definition 1 (Convex Cone & Conic Hull)** *A convex cone $\mathcal{C} \in \mathbb{R}^D$ is a convex set that is positively homogeneous: $\mathcal{C} = \tau \mathcal{C}$ for all $\tau > 0$. The convex conic hull of a sublevel set $S(\epsilon) := \{\mathbf{w} \in \mathbb{R}^D : \mathcal{L}(\mathbf{w}) \leq \mathcal{L}(\mathbf{w}^*) + \epsilon\}$ is:*

$$\mathcal{C}(S(\epsilon)) := \{\mathbf{w} \in \mathbb{R}^D : \mathcal{L}(\eta \mathbf{w}) \leq \mathcal{L}(\mathbf{w}^*) + \epsilon \quad \text{for some } \eta > 0\} \tag{5}$$

Statistical dimension is a useful metric to characterize the complexity of a convex cone. Intuitively speaking, the bigger the cone, the larger the statistical dimension, as illustrated in Fig. 1(b).

**Definition 2 (Statistical Dimension)** *The statistical dimension $\delta(\mathcal{C})$ of a convex cone $\mathcal{C}$ is:*

$$\delta(\mathcal{C}) := \mathbb{E}[\|\Pi_{\mathcal{C}}(\mathbf{g})\|_2^2] \tag{6}$$

*where $\Pi_{\mathcal{C}}$ is the Euclidean metric projector onto $\mathcal{C}$ and $\mathbf{g} \sim (\mathbf{0}, \mathbf{I}_{D \times D})$ is a standard normal vector.*

To characterize the sufficient and necessary condition of the set (or cone) intersection, the Approximate kinematics Formula Amelunxen et al. (2014) is a powerful and sharp result, which basically says that for two convex cones (or generally, sets), if the sum of their statistical dimension exceeds the ambient dimension, then these two cones would intersect with probability 1, otherwise they would intersect with probability 0.

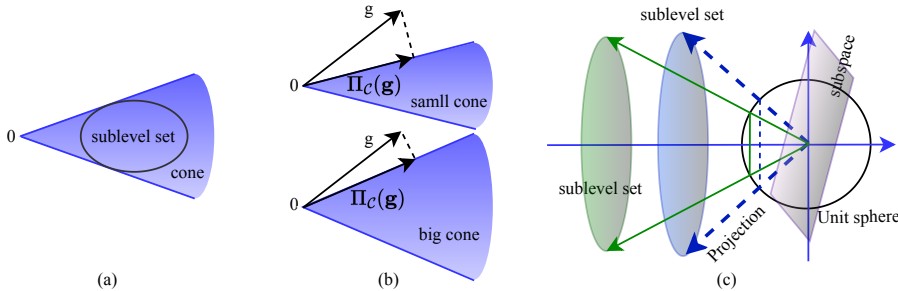

**Figure 1: Panel (a):** Illustration of a convex conic hull of a sublevel set. **Panel (b):** Illustration of the statistical dimension. **Panel (c):** Effect of projection distance on projection size and intersection probability.

**Theorem 1 (Approximate kinematics Formula)** *Let $\mathcal{C}$ be a convex conic hull of a sublevel set $S(\epsilon)$ in $\mathbb{R}^D$, and draw a random orthogonal basis $\mathbf{Q} \in \mathbb{R}^{D \times D}$. For a $k$-dimensional subspace $L_k$, it holds that (Amelunxen et al., 2014):*

$$\delta(\mathcal{C}) + k \lesssim D \rightarrow \mathbb{P}\{\mathcal{C} \cap \mathbf{Q}L_k = \{\mathbf{0}\}\} \approx 1$$
$$\delta(\mathcal{C}) + k \gtrsim D \rightarrow \mathbb{P}\{\mathcal{C} \cap \mathbf{Q}L_k = \{\mathbf{0}\}\} \approx 0 \tag{7}$$

Theorem 1 indicates that when $k \lesssim D - \delta(\mathcal{C})$, the $k$-dimensional subspace and the cone do not share a ray.

## 3 Lower Bound: No Sparse Solution Can be Found

In this section, we aim to characterize the lower bound of the pruning ratio, i.e., when the pruning ratio falls below a threshold, it's impossible to keep the generalization performance nearly unaffected. To establish the impossibility result, we'll leverage the powerful Approximate Kinematics Formula as detailed in Theorem 1

### 3.1 Network Pruning: Perspective from Set Intersection

**Sparse Space** Consider a $k$-dimensional sparse subspace contained in $D$ dimensional weight space, parameterized by $\theta \in \mathbb{R}^k : \mathbf{w}(\theta) = \mathbf{A}\theta + \mathbf{w}^k$. Here the columns of $\mathbf{A} \in \mathbb{R}^{D \times k}$ are random $k$ standard bases in $\mathbb{R}^D$. If $\mathbf{w}(\theta)$ and $S(\epsilon)$ do not intersect, $\mathbf{w}^k \notin S(\epsilon)$, there is no $k$-sparse solution.

In order to apply Theorem 1, we translate $\mathbf{w}(\theta)$ to the origin and perform a simultaneous translation of $S(\epsilon)$ in the same direction and distance, and we denote the translated $S(\epsilon)$ as $S_{\mathbf{w}^k}$. The non-existence of sparse solutions can be formulated as: $S_{\mathbf{w}^k} \cap \mathbf{A}\theta = \emptyset$, since $\mathbf{A}\theta$ is a subspace centered at the origin, $S_{\mathbf{w}^k} \cap \mathbf{A}\theta = \emptyset$ and $\mathcal{C}(S_{\mathbf{w}^k}) \cap \mathbf{A}\theta = \emptyset$ are equivalent. Therefore, we can utilize the statistical dimension in theorem 1 to determine the threshold where $d$-sparse solutions do not exist. Next, we will introduce the Gaussian width for calculating the statistical dimension of $\mathcal{C}(S_{\mathbf{w}^k})$.

**Definition 3 (Gaussian Width)** . *The gaussian width of a subset $S \in \mathbb{R}^D$ is given by:*

$$w(S) = \frac{1}{2}\mathbb{E} \sup_{\mathbf{x}, \mathbf{y} \in S} \langle \mathbf{g}, \mathbf{x} - \mathbf{y} \rangle, \mathbf{g} \sim \mathcal{N}(\mathbf{0}, \mathbf{I}_{D \times D}). \tag{8}$$

Amelunxen et al. (2014) indicates that the Gaussian width of a spherical convex set is comparable with the statistical dimension of the cone generated by the set:

**Theorem 2** *Given a unit sphere $\mathbb{S}^{D-1} := \{\mathbf{x} \in \mathbb{R}^D : \|\mathbf{x}\| = 1\}$, let $\mathcal{C}$ be a convex cone in $\mathbb{R}^D$, then:*

$$w(\mathcal{C} \cap \mathbb{S}^{D-1})^2 \leq \delta(\mathcal{C}) \leq w(\mathcal{C} \cap \mathbb{S}^{D-1})^2 + 1 \tag{9}$$

As theorem 2 requires the set to be a spherical convex set, for using Gaussian width as a proxy for statistical dimension, we need project the sublevel set $S_{\mathbf{w}^k}$ onto the surface of the unit sphere centered at origin. The projection of $S_{\mathbf{w}^k}$ is defined as:

$$\text{proj}(S_{\mathbf{w}^k}) = \{(\mathbf{x} - \mathbf{w}^k)/\|\mathbf{x} - \mathbf{w}^k\|_2, \mathbf{x} \in S\} \tag{10}$$

Consider two manifolds with equal width, it can be observed that as the distance from the sphere increases, the projected size on the sphere decreases, the Gaussian width of the spherical convex set also decreases, leading to the cone becoming smaller with a reduced probability of intersection with the subspace. This relationship is visually depicted in Figure 1(c). Under the projection setting, theorem 1 of the pruning work is adjusted to:

**Theorem 3 (Network Pruning Approximate Kinematic)** *Let $\mathcal{C}$ be a convex conic hull of a sublevel set $S_{\mathbf{w}^k}$ in $\mathbb{R}^D$. For a $k$-dimensional subspace $\mathbf{A}\boldsymbol{\theta}$ and draw a random orthogonal basis $\mathbf{Q} \in \mathbb{R}^{D \times D}$, it holds that:*

$$w(proj(S_{\mathbf{w}^k}))^2 + k \lesssim D \to \mathbb{P}\{\mathcal{C}(S_{\mathbf{w}^k}) \cap \mathbf{Q}\mathbf{A}\boldsymbol{\theta} = \{\mathbf{0}\}\} \approx 1$$
$$w(proj(S_{\mathbf{w}^k}))^2 + k \gtrsim D \to \mathbb{P}\{\mathcal{C}(S_{\mathbf{w}^k}) \cap \mathbf{Q}\mathbf{A}\boldsymbol{\theta} = \{\mathbf{0}\}\} \approx 0 \tag{11}$$

This theorem tells us that when the dimension of the sub-network is lower than $D - w(proj(S_{\mathbf{w}^k}))^2$, the subspace will not intersect with $S_{\mathbf{w}^k}$, ie., no sparse solution can be found. Therefore, the lower bound of the pruning ratio of the network $M$ can be expressed as:

$$T(M, \mathbf{w}^k) = \frac{D - w(\text{proj}_{\mathbf{w}^k}(S))^2}{D} = 1 - \frac{w(\text{proj}_{\mathbf{w}^k}(S))^2}{D}. \tag{12}$$

## 3.2 Reformulation of The Sublevel Set

Consider a well-trained deep neural network model $M^*$ with weights $\mathbf{w}^*$ and an arbitrary loss function $\mathcal{L}(\mathbf{w})$, where $\mathbf{w}$ lies in a small neighborhood of $\mathbf{w}^*$. Perform a Taylor expansion of $\mathcal{L}(\mathbf{w})$ at $\mathbf{w}^*$:

$$\mathcal{L}(\mathbf{w}) = \mathcal{L}(\mathbf{w}^*) + (\mathbf{w} - \mathbf{w}^*)\mathbf{G} + \frac{1}{2}(\mathbf{w} - \mathbf{w}^*)^T\mathbf{H}(\mathbf{w} - \mathbf{w}^*) + \Delta. \tag{13}$$

where $\mathbf{G}$ and $\mathbf{H}$ denote the first and second derivatives of $\mathcal{L}(\mathbf{w})$ with respect to the model parameters $\mathbf{w}$, and $\Delta$ represents the higher order terms in the Taylor expansion which can be ignored.
For a well-trained deep neural network model, the first derivatives of $\mathcal{L}(\mathbf{w})$ satisfy $\mathbf{G} = \mathbf{0}$ and the second derivatives $\mathbf{H}$ is a positive definite matrix. Consequently, the loss sublevel set $S(\epsilon)$ can be expressed as:

$$S(\epsilon, \mathbf{w}^*) = \{\hat{\mathbf{w}} \in \mathbb{R}^D : \frac{1}{2}\hat{\mathbf{w}}^T\mathbf{H}\hat{\mathbf{w}} \le \epsilon\} \tag{14}$$

where $\hat{\mathbf{w}} = \mathbf{w} - \mathbf{w}^*$. Due to the positive definiteness property of $\mathbf{H}$, $S(\epsilon, \mathbf{w}^*)$ forms an ellipsoid, and the proof regarding the ellipsoid can be found in Appendix C.1.

## 3.3 Gaussain Width of the Ellipsoid

We leverage tools in high-dimensional probability, especially the concentration of measure, which enables us to present a rather precise expression for the Gaussian width of the high-dimensional ellipsoid.

**Lemma 1** *Give an ellipsoid $S(\epsilon)$ defined by a quadratic form: $S(\epsilon) := \{\mathbf{w} \in \mathbb{R}^D : \frac{1}{2}\mathbf{w}^T\mathbf{H}\mathbf{w} \le \epsilon\}$ where $\mathbf{H} \in \mathbb{R}^{D \times D}$ is a symmetric, positive definite Hessian matrix. Loss sublevel set $S(\epsilon)$ defined by $\mathbf{H}$ is an ellipsoidal body with the Gaussian width:*

$$w(S(\epsilon)) \approx (2\epsilon \text{Tr}(\mathbf{H}^{-1}))^{1/2} = (\sum_i r_i^2)^{1/2} \tag{15}$$

*where $r_i = \sqrt{2\epsilon/\lambda_i}$ is the radius of ellipsoidal body and $\lambda_i$ is the $i$-th eigenvalue of $\mathbf{H}$.*

The proof of Lemma 1 is in Appendix C.1. We next perform translation and projection operation to the loss sublevel set $S(\epsilon, \mathbf{w}^*)$:

**Lemma 2** *Consider a projection $proj_{\mathbf{w}^k}(S)$ defined in Eq.(10) and the projection distance $R = \|\mathbf{w}^* - \mathbf{w}^k\|_2$. Larsen et al. (2021) outline the Gaussian width of the projected quadratic well is adjusted to:*

$$w(proj_{\mathbf{w}^k}(S)) = (\sum_i \frac{r_i^2}{R^2 + r_i^2})^{1/2} \tag{16}$$

Therefore, the projected Gaussian width of $S(\epsilon, \mathbf{w}^*)$ defined in Eq.(14) becomes:

$$w(\text{proj}_{\mathbf{w}^k}(S)) = (\sum_i \frac{r_i^2}{R^2 + r_i^2})^{1/2} \quad \text{with } r_i = \sqrt{2\epsilon/\lambda_i} \tag{17}$$

The Gaussian width of a projected ellipsoid demonstrated by Larsen et al. (2021) is $[(\sqrt{\frac{2}{\pi}}\sum_i \frac{r_i^2}{R^2+r_i^2})^{1/2}, (\sum_i \frac{r_i^2}{R^2+r_i^2})^{1/2}]$, the most notable **difference** is that our established Gaussian width $(\sum_i \frac{r_i^2}{R^2+r_i^2})^{1/2}$ is not represented as an interval but as a precise and definite value.

### 3.4 LOWER BOUND OF NETWORK PRUNING RATIO

According to Eq.(12), we have the lower bound of the pruning ratio as follows:

**Corollary 1** *For a well-trained deep neural network model $M$ with weights $\mathbf{w} \in \mathbb{R}^D$ and an arbitrary loss function $\mathcal{L}(\mathbf{w})$, the lower bound of pruning ratio of model $M$ is:*

$$T(M, \mathbf{w}^k) = \frac{D - w(proj_{\mathbf{w}^k}(S))^2}{D} = 1 - \frac{1}{D}\sum_i \frac{r_i^2}{R^2+r_i^2} \quad with \; r_i = \sqrt{2\epsilon/\lambda_i}. \tag{18}$$

*where $\lambda_i$ is the eigenvalue of the Hessian matrix of the loss function $\mathcal{L}(\mathbf{w})$ with respect to $\mathbf{w}$ and $R$ is the projection distance: $R = \|\mathbf{w}^* - \mathbf{w}^k\|_2$.*

## 4 UPPER BOUND: THE EXISTENCE OF SPARSE SOLUTION

In order to establish the upper bound of the pruning ratio, the main task is to prove that the $m$-sparse weight vector after pruning intersects with the loss sub-level set. To achieve that aim, we leverage the fact that according to our proposed magnitude-based pruning procedure, all the erased weights are of small magnitudes, therefore, the sub-vector comprised by them is very near to the origin, thus intuitively speaking, the origin is very likely within the loss sub-level set centered by that sub-vector. This heuristic reasoning can be made rigorous by exploiting the statistical dimension framework and the Approximate Kinematic Formula (Amelunxen et al., 2014).

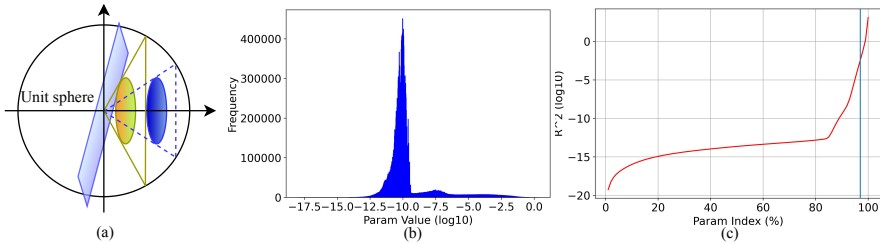

Figure 2: Effect of extremely small projection distance on projection size and intersection probability and statistical information of ResNet50 on TinyImagenet. Statistical information of all experiments can be found in Appendix E.

Given the weights $\mathbf{w}^*$ which is arranged in descending order, we categorize $\mathbf{w}^*$ into two distinct part: $\mathbf{w}^1 = [\mathbf{w}_1^*, \mathbf{w}_2^*, \ldots, \mathbf{w}_m^*]$, comprising the $m$ largest parameters, and $\mathbf{w}^2 = [\mathbf{w}_{m+1}^*, \mathbf{w}_{d+2}^*, \ldots, \mathbf{w}_D^*]$, encompassing the remaining $D - m$ parameters. Fixing $\mathbf{w}^1$ and taking a specific focus on $\mathbf{w}^2$, we establish the definition of the loss sublevel set denoted as:

$$S(\mathbf{w}^2) = \{\mathbf{w}^2 \in \mathbb{R}^{D-m} : \mathcal{L}([\mathbf{w}^1, \mathbf{w}^2]) \leq \mathcal{L}(\mathbf{w}^*) + \epsilon\}$$

In the scenario where $S(\mathbf{w}^2)$ intersects with any subspace of any dimension, it implies that the origin is within $S(\mathbf{w}^2)$. Consequently, the existence of an $m$-dimensional sparse solution is confirmed.

The sublevel set $S(\mathbf{w}^2)$ remains an ellipsoid, as demonstrated in Appendix D.1. When projecting $S(\mathbf{w}^2)$, centered at $\mathbf{w}^2$, onto the surface of the unit sphere with the projection distance $R_m = \|\mathbf{w}^2\|_2$, as per Lemma 2, the resulting Gaussian width of the projected $S(\mathbf{w}^2)$ is $(\sum_i^{D-m} \frac{r_i^2}{R_m^2+r_i^2})^{1/2}$ with $r_i = \sqrt{2\epsilon/\lambda_i}$, where $\lambda_i$ is the eigenvalue of the hessian matrix of $\mathcal{L}([\mathbf{w}^1, \mathbf{w}^2])$ with respect to $\mathbf{w}^2$, a subspace of dimension $k = D - m - \sum_i^{D-m} \frac{r_i^2}{R_m^2+r_i^2}$ is required for intersection with $S(\mathbf{w}^2)$. Therefore, the minimal of $m$ which obeys $D - m - \sum_i \frac{r_i^2}{R_m^2+r_i^2} = 0$ is the lower bound of pruning threshold, at this pruning point, all elements in $\mathbf{w}^2$ can be pruned with $\mathbf{w}^1 \in \mathbb{R}^m$ remains. Hence, the upper bound for the pruning ratio is the fix-point of $U(p)$ where $p = m/D$:

$$U(p) = (D - \sum_i^{D-m} \frac{r_i^2}{R_m^2+r_i^2})/D = 1 - \frac{1}{D}\sum_i^{D-m} \frac{r_i^2}{R_m^2+r_i^2} \tag{19}$$

Training based on $l_1$-regularization leads to the presence of a substantial number of weights with vanishingly small magnitudes in the network. As shown in Figure 2(b), the majority of weights are extremely small, resulting in a very small projection distance $\mathbf{w}^2$. The curve in 2(c) represents $R_m^2$ with respect to $m/D$, which is the sum of squares of the $D - m$ smaller parameters in $\mathbf{w}^*$. The vertical line represents $T$, which is the lower bound of the pruning ratio predicted in Section 3. When $m = DT$, as the weights in $\mathbf{w}^2$ are all very small, the square of the projection distance $R_m$ is also small, which can be confirmed by the intersection point in Figure 2(c). Therefore, the upper bound for the pruning ratio is:

$$U(p) = 1 - \frac{1}{D} \sum_i^{D-m} \frac{r_i^2}{R_m^2 + r_i^2} \approx 1 - \frac{1}{D} \sum_i^{D-m} \frac{r_i^2}{r_i^2} = 1 - \frac{D - m}{D} = 1 - \frac{D - DT}{D} = T \quad (20)$$

Eq. 20 indicates that the upper bound of the pruning ratio is approximately equal to the lower bound. Here, we provide experimental statistical information. We independently replicated the experiments five times across eight tasks and recorded the differences between the upper bound and lower bound, denoted as $\Delta$, the statistical information is shown in Table 1.

Table 1: The Difference Between Lower Bound and Upper Bound of Pruning Ratio.

| CIFAR10 | FC5 | FC12 | Alexnet | VGG16 |
|---|---|---|---|---|
| $\Delta$ | 0.25%±0.09% | 0.03%±0.02% | 0.02%±0.01% | 0.01%±0% |
| **ResNet** | **18 on CIFARF100** | **50 on CIFARF100** | **18 on TinyImagenet** | **50 on TinyImagenet** |
| $\Delta$ | 0.01%±0.01% | 0.01%±0.01% | 0.31%±0.08% | 0.27%±0.11% |

**Computational Challenges & Countermeasures**: In practice, determining the Gaussian width of the ellipsoid defined by the network loss function is a challenging task. The network normally fails to converge perfectly to its extremum, leading to a non-positive definite Hessian matrix for the loss function, thus deforming the original ellipsoid. To address this problem, we instead calculate the convex hull of non-convex body resulting from non-positive definite matrices. The proof that the convexifying processing has no impact on the Gaussian width is presented in Appendix C.2.

Furthermore, neural networks often exhibit a significant number of zero or exceedingly small eigenvalues in their Hessian matrices. It's thus hard for the spectrum estimation algorithm SLQ(Stochastic Lanczos Quadrature) proposed by Yao et al. (2020) to accurately estimate these eigenvalues. To address this issue, we enhance the existing large-scale spectrum estimation algorithms by a key modification, i,e, to estimate the number of these exceptionally small eigenvalues by employing Hessian matrix sampling . A comprehensive algorithm description and the implemented improvements are presented in Appendix B.

**Final Result**: As is shown in Table 1, the upper bound is approximately equal to the lower bound, thus we can use **the lower bound** in corollary 1 as the final **pruning ratio threshold**.

## 5 EXPERIMENTS

In this section, we experimentally validate our pruning method and lower bound on network pruning using the Approximate Kinematics Formula.

**Tasks.** We evaluate the pruning algorithm and pruning ratio threshold on: Full-Connect-5(FC5), Full-Connect-12(FC12), AlexNet (Krizhevsky et al., 2017) and VGG16 (Simonyan & Zisserman, 2014) on CIFAR10 (Krizhevsky et al., 2009), ResNet18 and ResNet50 (He et al., 2016) on CIFAR100 and TinyImageNet (Le & Yang, 2015). We make use of theoretical principles to anticipate the pruning ratio limit of the network, followed by an evaluation of the sparse sub-networks performance at different sparse ratios on test data. Specifically, we calculate accuracy and loss metrics to quantify their performance. Finally, we compare the predicted lower bound on the pruning ratio with the actual pruning ratio and evaluate whether they match. Detailed descriptions of datasets, networks, hyper-parameters and eigenspectrum adjustment can be found in Section A of the Appendix.

### 5.1 PRUNING LOWER BOUND: VALIDATION

We validated our prediction results on all tasks for the lower bound of network pruning. The images generated during the experimental process, such as the statistical plots of Hessian matrix row $l_1$ norm, can be found in Appendix B. Figure. 3 shows the sparsity-accuracy trade-off in all tasks and provides compelling evidence of the high redundancy of deep neural networks. The theoretical lower bound we derive matches well with the practical pruning ratio threshold.

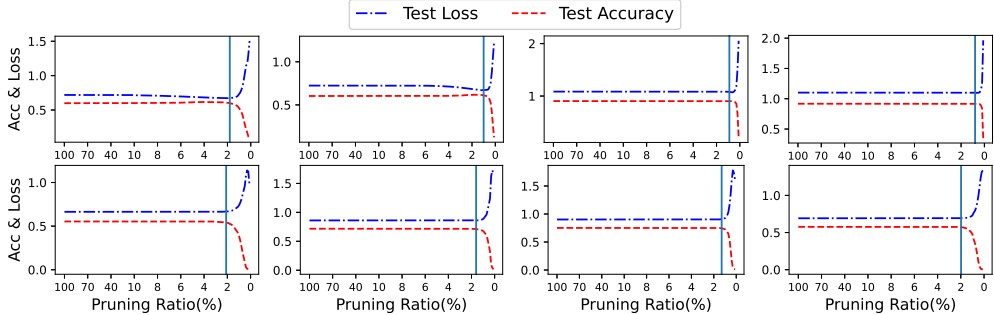

Figure 3: We assessed the influence of sparsity on loss and test accuracy using the test dataset, and we marked the theoretical pruning ratio with vertical lines. The first row, from left to right, corresponds to FC5, FC12, AlexNet, and VGG16. The second row, from left to right, corresponds to ResNet18 and ResNet50 on CIFAR100, as well as ResNet18 and ResNet50 on TinyImagenet. The figures demonstrate that our theory predicts the critical pruning points quite accurately.

## 5.2 PREDICTION COMPARISON

The numerical comparison between the predicted pruned weights ratio and the actual value is shown in Table 2. The results in Table 2 exhibit a high degree of agreement between the predicted and actual values which demonstrate that our theoretical predictions effectively estimate the network pruning ratio threshold.

Table 2: Comparison between Prediction of Pruned Parameters Ratio and Actual Values.

| Dataset | Model | Prediction(%) | Experimental Results(%) | $\Delta$(%) |
|---------|-------|---------------|-------------------------|-------------|
| CIFAR10 | FC5 | 97.9±0.25 | 98.3±0.12 | -0.40±0.35 |
| | FC12 | 99.0±0.30 | 99.2±0.06 | -0.15±0.26 |
| | AlexNet | 99.1±0.00 | 99.2±0.08 | -0.14±0.08 |
| | VGG16 | 99.2±0.06 | 99.2±0.06 | 0.04±0.08 |
| CIFAR100 | ResNet18 | 98.5±0.05 | 98.0±0.13 | 0.54±0.15 |
| | ResNet50 | 98.1±0.05 | 97.9±0.16 | 0.28±0.19 |
| TinyImageNet | ResNet18 | 96.1±0.82 | 95.7±0.38 | 0.46±0.71 |
| | ResNet50 | 97.4±0.24 | 97.1±0.33 | 0.36±0.10 |

## 5.3 COMPARISON OF PRUNING PERFORMANCE

We validated $l_1$-regularization based global one-shot pruning algorithm(GOP) against four baselines: dense weight training and three pruning algorithms: (i) Rare Gems(RG) proposed by Sreenivasan et al. (2022), (ii) Lottery Tickets Hypolothis(LTH) donated by Frankle & Carbin (2018), (iii) Smart-Ratio (SR) which is the random pruning method proposed by Su et al. (2020). Table 3 shows the pruning performance of the above algorithms, our pruning algorithm is better performing than other algorithms.

Table 3: Performance comparison of various pruning algorithms.

| Dataset | Model | Dense Acc (%) | Sparsity (%) | Test Acc (%)@top-1 | | | |
|---------|-------|---------------|--------------|--------------------|-----|-----|-----|
| | | | | GOP(ours) | RG | LTH | SR |
| CIFAR10 | FC5 | 55.3±0.62 | 1.7 | **59.96±0.45** | 58.76±0.15 | 38.71±2.25 | - |
| | FC12 | 55.5±0.26 | 1.0 | **60.84±0.21** | 54.96±0.28 | 10.00±0.00 | - |
| | AlexNet | 89.60±0.31 | 0.7 | **90.55±0.04** | 85.55±0.11 | 21.65±2.63 | 10.00±0.00 |
| | VGG16 | 90.73±0.22 | 0.6 | **91.66±0.08** | 87.66±0.11 | 86.59±0.21 | 10.00±0.00 |
| CIFAR100 | ResNet18 | 72.19±0.23 | 1.9 | **71.82±0.09** | 67.52±0.30 | 64.42±0.20 | 66.50±0.24 |
| | ResNet50 | 74.07±0.43 | 2.0 | **75.22±0.11** | 70.96±0.23 | 61.17±0.45 | 66.99±0.21 |
| TinyImageNet | ResNet18 | 52.92±0.13 | 4.2 | **55.42±0.02** | 37.14±0.20 | 55.02±0.27 | 53.28±0.25 |
| | ResNet50 | 56.45±0.17 | 2.9 | **57.49±0.01** | 36.78±0.13 | 49.35±1.27 | 53.65±0.34 |

## 6 DISCUSSION

**Iteration is needed in IMP.** In the IMP (Iterative Magnitude Pruning) work, we determine the pruning ratio thresholds for various stages through calculations, as depicted in the top row of Figure 4. It is noteworthy that the pruning rate threshold at each stage does not directly support one-shot pruning to the final sparsity level, necessitating multiple iterations of pruning under IMP's

configuration. As the pruning depth gradually increases, the theoretical pruning ratio threshold also increases. Therefore, it is appropriate to prune smaller proportions of weights gradually during iterative pruning, in this context, the pruning rate refers to the proportion of retained weights in the current active weights (chosen by a mask) during this pruning operation. Both Zhu & Gupta (2017) and Sreenivasan et al. (2022) have employed pruning rate adjustment algorithms, gradually pruning smaller proportions of the weights with the iteration of the algorithm.

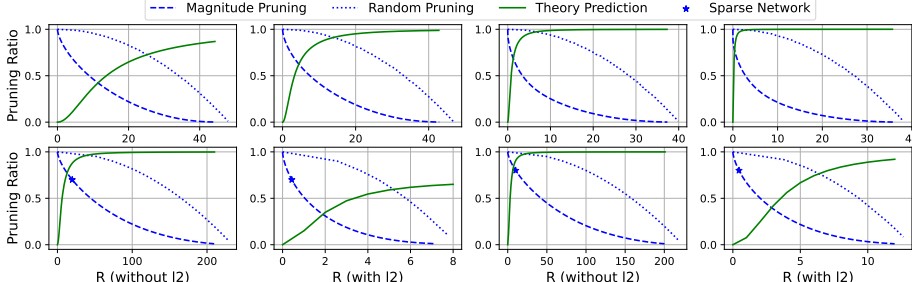

Figure 4: **Top Row:** The theoretical pruning ratio threshold in IMP of ResNet50 on TinyImagenet, respectively. From left to right, as the number of iterations increases, it leads to an increase in the theoretical pruning ratio threshold. **Bottom Row:** The comparison of the pruning ratio threshold in pruning for ResNet50 on TinyImagenet when using and not using $l_2$-regularization. Sparse networks are obtained by magnitude-based pruning with fixed pruning ratios. The two plots on the left and the two plots on the right correspond to different fixed pruning ratios.

$l_2$**-regularization enhances the performance of Rare Gems.** In Rare Gems, Sreenivasan et al. (2022) shows that the use of $l_2$ regularization and its absence led to significant differences in the final performance, we have similarly scrutinized the differences between these approaches during pruning as is shown in the bottom row of Figure 4. We have discovered that when $l_2$-regularization is applied, the pruning ratios tend to be larger than the theoretical limits, whereas the absence of $l_2$-regularization results in excessive pruning, which can be regarded as wrong pruning.

**Magnitude pruning is better than random pruning.** In Figure 4, we depict the curves for random pruning. Under the same pruning ratio (For instance, in the top-left subplot of Figure 4, focusing on a pruning ratio of 0.5.), when using random pruning, the pruning occurs below the theoretically predicted curve, indicating the absence of current sparse solutions. Therefore, the heuristic magnitude pruning performs better than random pruning. The influence of weight magnitudes on the pruning ratio threshold can be found in Appendix E.1, and its brief description is as follows: Smaller magnitudes result in a lower pruning ratio threshold.

## 7 CONCLUSION

In this paper we explore the fundamental limit of pruning ratio of deep networks by utilizing the framework of high dimensional geometry, thus, the pruning limit problem can be reduced to determine whether two sets (cones, subspaces etc.) intersect. Through this geometric perspective, powerful tools, such as statistical dimension and Approximate kinematic formula, can be leveraged. Thus we can for the first time characterize the *sharp* phase transition point of network pruning (namely, the pruning ratio threshold), and moreover, with a very succinct form. The key message is that the fundamental limit of network pruning is mostly determined by the magnitude of the weight vector as well as the spectrum of the Hessian matrix corresponding to the weight vector. Equipped with this guidelines, we're able to provide clear explanations of many phenomenon of existing pruning algorithms. Furthermore, to address the challenges in computing the associated Gaussian width, we develop an improved spectrum estimation for large Hessian matrices. Experiments demonstrate both the high accuracy of our theoretical result and the excellent performance of our proposed pruning algorithm.

**Future work.** Inspired by the fact that the maximal pruning ratio of a DNN is succinctly determined by the statistical dimension of a given cone (induced by the loss landscape), it's natural to ask: whether that statistical dimension can be served as a measure of the capacity of the DNN? In another regard, the pruning procedures considered in our paper is based on the result after the training, an intriguing problem is: whether is it possible to prune the network before the training, and what's its fundamental limit?

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
