# How Sparse Can We Prune A Deep Network: A Geometric Viewpoint

## A    Experimental Details

In this section, we describe the datasets, models, hyper-parameter choices and eigenspectrum adjustment used in our experiments. All of our experiments are run using PyTorch 1.12.1 on Nvidia RTX3090s with ubuntu20.04-cuda11.3.1-cudnn8 docker.

### A.1    Dataset

**CIFAR-10.**    CIFAR-10 consists of 60,000 color images, with each image belonging to one of ten different classes with size $32 \times 32$. The classes include common objects such as airplanes, automobiles, birds, cats, deer, dogs, frogs, horses, ships, and trucks. The CIFAR-10 dataset is divided into two subsets: a training set and a test set. The training set contains 50,000 images, while the test set contains 10,000 images (Krizhevsky et al., 2009). For data processing, we follow the standard augmentation: normalize channel-wise, randomly horizontally flip, and random cropping.

**CIFAR-100.**    The CIFAR-100 dataset consists of 60,000 color images, with each image belonging to one of 100 different fine-grained classes (Krizhevsky et al., 2009). These classes are organized into 20 superclasses, each containing 5 fine-grained classes. Similar to CIFAR-10, the CIFAR-100 dataset is split into a training set and a test set. The training set contains 50,000 images, and the test set contains 10,000 images. Each image is of size 32x32 pixels and is labeled with its corresponding fine-grained class. Augmentation includes normalize channel-wise, randomly horizontally flip, and random cropping.

**TinyImageNet.**    TinyImageNet comprises 100,000 images distributed across 200 classes, with each class consisting of 500 images (Le & Yang, 2015). These images have been resized to 64 × 64 pixels and are in full color. Each class encompasses 500 training images, 50 validation images, and 50 test images. Data augmentation techniques encompass normalization, random rotation, and random flipping. The dataset includes distinct train, validation, and test sets for experimentation. For data preprocessing, please refer to Lokesh (2020).

### A.2    Model

In all experiments, pruning skips bias and batchnorm, which have little effect on the sparsity of the network. Use non-affine batchnorm in the network, and the initialization of the network is kaiming normal initialization.

**Full Connect Network(FC-5, FC-12).**    We train a five-layer fully connected network (FC-5) and a twelve-layer fully connected network FC-12 on CIFAR-10, the network architecture details can be found in Table 1.

Table 1: FC-5 and FC-12 architecture used in our experiments.

| Model | Layer Width |
|---|---|
| FC-5 | 1000, 600, 300, 100, 10 |
| FC-12 | 1000, 900, 800, 750, 700, 650, 600, 500, 400, 200, 100, 10 |

Table 2: AlexNet architecture used in our experiments.

| Layer | Shape | Stride | Padding |
|---|---|---|---|
| conv1 | $3 \times 96 \times 11 \times 11$ | 4 | 1 |
| max pooling | kernel size:3 | 2 | N/A |
| conv2 | $96 \times 256 \times 5 \times 5$ | 1 | 2 |
| max pooling | kernel size:3 | 2 | N/A |
| conv3 | $256 \times 384 \times 3 \times 3$ | 1 | 1 |
| conv4 | $384 \times 384 \times 3 \times 3$ | 1 | 1 |
| conv4 | $384 \times 256 \times 3 \times 3$ | 1 | 1 |
| max pooling | kernel size:3 | 2 | N/A |
| linear1 | $6400 \times 4096$ | N/A | N/A |
| linear1 | $4096 \times 4096$ | N/A | N/A |
| linear1 | $4096 \times 10$ | N/A | N/A |

Table 3: VGG-16 architecture used in our experiments.

| Layer | Shape | Stride | Padding |
|---|---|---|---|
| conv1 | $3 \times 64 \times 3 \times 3$ | 1 | 1 |
| conv2 | $64 \times 64 \times 3 \times 3$ | 1 | 1 |
| max pooling | kernel size:2 | 2 | N/A |
| conv3 | $64 \times 128 \times 3 \times 3$ | 1 | 1 |
| conv4 | $128 \times 128 \times 3 \times 3$ | 1 | 1 |
| max pooling | kernel size:2 | 2 | N/A |
| conv5 | $128 \times 256 \times 3 \times 3$ | 1 | 1 |
| conv6 | $256 \times 256 \times 3 \times 3$ | 1 | 1 |
| conv7 | $256 \times 256 \times 3 \times 3$ | 1 | 1 |
| max pooling | kernel size:2 | 2 | N/A |
| conv8 | $256 \times 512 \times 3 \times 3$ | 1 | 1 |
| conv9 | $512 \times 512 \times 3 \times 3$ | 1 | 1 |
| conv10 | $512 \times 512 \times 3 \times 3$ | 1 | 1 |
| max pooling | kernel size:2 | 2 | N/A |
| conv11 | $512 \times 512 \times 3 \times 3$ | 1 | 1 |
| conv12 | $512 \times 512 \times 3 \times 3$ | 1 | 1 |
| conv13 | $512 \times 512 \times 3 \times 3$ | 1 | 1 |
| max pooling | kernel size:2 | 2 | N/A |
| avg pooling | kernel size:1 | 1 | N/A |
| linear1 | $512 \times 10$ | N/A | N/A |

**AlexNet (Krizhevsky et al., 2017).** We use the standard AlexNet architecture. In order to use CIFAR-10 to train AlexNet, we upsample each picture of CIFAR-10 to $3 \times 224 \times 224$. The detailed network architecture parameters are shown in Table 2.

**VGG-16 (Simonyan & Zisserman, 2014).** In the original VGG-16 network, there are 13 convolution layers and 3 FC layers (including the last linear classification layer). We follow the VGG-16 architectures used in Frankle et al. (2020a;b) to remove the first two FC layers while keeping the last linear classification layer. This finally leads to a 14-layer architecture, but we still call it VGG-16 as it is modified from the original VGG-16 architectural design. Detailed architecture is shown in Table 3. VGG-16 is trained on CIFAR-10.

**ResNet-18 and ResNet-50 (He et al., 2016).** We use the standard ResNet architecture for Tiny-ImageNet and tune it for the CIFAR-100 dataset. The detailed network architecture parameters are shown in Table 4. ResNet-18 and ResNet-50 is trained on CIFAR-100 and TinyImageNet.

Table 4: ResNet architecture used in our experiments.

| Layer | ResNet-18 | ResNet-50 |
|---|---|---|
| conv1 | $64, 3 \times 3$; stride:1; padding:1 | $64, 3 \times 3$; stride:1; padding:1 |
| block1 | $\begin{pmatrix} 64, 3 \times 3; \text{stride:1; padding:1} \\ 64, 3 \times 3; \text{stride:1; padding:1} \end{pmatrix} \times 2$ | $\begin{pmatrix} 64, 1 \times 1; \text{stride:1; padding:0} \\ 64, 3 \times 3; \text{stride:1; padding:1} \\ 256, 1 \times 1; \text{stride:1; padding:0} \end{pmatrix} \times 3$ |
| block1 | $\begin{pmatrix} 128, 3 \times 3; \text{stride:2; padding:1} \\ 128, 3 \times 3; \text{stride:1; padding:1} \end{pmatrix} \times 2$ | $\begin{pmatrix} 128, 1 \times 1; \text{stride:1; padding:0} \\ 128, 3 \times 3; \text{stride:2; padding:1} \\ 512, 3 \times 3; \text{stride:1; padding:0} \end{pmatrix} \times 4$ |
| block1 | $\begin{pmatrix} 128, 3 \times 3; \text{stride:2; padding:1} \\ 256, 3 \times 3; \text{stride:1; padding:1} \end{pmatrix} \times 2$ | $\begin{pmatrix} 256, 1 \times 1; \text{stride:1; padding:0} \\ 256, 3 \times 3; \text{stride:2; padding:1} \\ 1024, 1 \times 1; \text{stride:1; padding:0} \end{pmatrix} \times 6$ |
| block1 | $\begin{pmatrix} 512, 3 \times 3; \text{stride:2; padding:1} \\ 512, 3 \times 3; \text{stride:1; padding:0} \end{pmatrix} \times 2$ | $\begin{pmatrix} 512, 1 \times 1; \text{stride:1; padding:1} \\ 512, 3 \times 3; \text{stride:2; padding:1} \\ 2048, 1 \times 1; \text{stride:1; padding:0} \end{pmatrix} \times 3$ |
| avg pooling | kernel size:1; stride:1 | kernel size:1; stride:1 |
| linear1 | $512 \times ClassNum$ | $2048 \times ClassNum$ |

### A.3 TRAIN HYPER-PARAMETER SETUP

In this section, we will describe in detail the training hyper-parameters of the Global One-shot Pruning algorithm on multiple datasets and models. The various hyperparameters are detailed in Table 5.

Table 5: Hyper Parameters used for different Datasets and Models.

| Model | Dataset | Batch Size | Epochs | Optimizer | LR | Momentum | Warm Up | Weight Decay | CosineLR | Lambda |
|---|---|---|---|---|---|---|---|---|---|---|
| FC5 | CIFAR10 | 128 | 200 | SGD | 0.01 | 0.9 | 0 | 0 | N/A | 0.00005 |
| FC12 | CIFAR10 | 128 | 200 | SGD | 0.01 | 0.9 | 0 | 0 | N/A | 0.00005 |
| VGG16 | CIFAR10 | 128 | 200 | SGD | 0.01 | 0.9 | 5 | 0 | True | 0.00015 |
| AlexNet | CIFAR10 | 128 | 200 | SGD | 0.01 | 0.9 | 5 | 0 | True | 0.00003 |
| ResNet18 | CIFAR100 | 128 | 200 | SGD | 0.1 | 0.9 | 5 | 0 | True | 0.000055 |
| ResNet50 | CIFAR100 | 128 | 200 | SGD | 0.1 | 0.9 | 5 | 0 | True | 0.00002 |
| ResNet18 | TinyImageNet | 128 | 200 | SGD | 0.01 | 0.9 | 5 | 0 | True | 0.00023 |
| ResNet50 | TinyImageNet | 128 | 200 | SGD | 0.01 | 0.9 | 5 | 0 | True | 0.0001 |

### A.4 SUBLEVEL SET PARAMETER SETUP.

Given that the test data is often unavailable and the network can ensure the Hessian matrix is positive definite as much as possible by utilizing the training data for computation. Additionally, we generally assume that the training and test data share the same distribution, thus we use the training data to define the loss sublevel set as $\hat{\epsilon} = \epsilon - \mathcal{L}(\mathbf{w}_0)$. We compute the standard deviation of the network's loss across multiple batches on the training data set and denote it by $\hat{\epsilon}$.

Table 6: Hyper Parameters used in SLQ Algorithm.

| Model | Dataset | Runs | Iterations | Bins | Squared Sigma |
|---|---|---|---|---|---|
| FC5 | CIFAR10 | 1 | 128 | 100000 | 1e-10 |
| FC12 | CIFAR10 | 1 | 128 | 100000 | 1e-10 |
| VGG16 | CIFAR10 | 1 | 128 | 100000 | 1e-07 |
| AlexNet | CIFAR10 | 1 | 96 | 100000 | 1e-07 |
| ResNet18 | CIFAR100 | 1 | 128 | 100000 | 1e-07 |
| ResNet50 | CIFAR100 | 1 | 128 | 100000 | 1e-07 |
| ResNet18 | TinyImageNet | 1 | 128 | 100000 | 1e-07 |
| ResNet50 | TinyImageNet | 1 | 88 | 100000 | 1e-07 |

### A.5 LOSS SUBLEVEL SET

Given a dense well-trained neural network $M$ with weighted donated as $\mathbf{w}^*$, the loss sublevel set is $\{\mathbf{w} \in \mathbb{R}^D : \frac{1}{2}\hat{\mathbf{w}}^T \mathbf{H}\hat{\mathbf{w}} \leq \hat{\epsilon}\}$ where $\hat{\epsilon} = \epsilon - \mathcal{L}(\mathbf{w}^*)$, as we operate under the assumption of having access only to training data, we calculate the training loss for each batch and utilize the standard deviation of all training losses as the variable $\hat{\epsilon}$.

### A.6 THEORETICALLY PREDICTED PRUNING RATIO

Taking $\mathbf{w}^*$ as the initial pruning point and calculating the corresponding value of $R$ for different pruning ratios. We then plot the corresponding curve of the theoretically predicted pruning ratio and the calculated $R$ in the same graph. The intersection point of these two curves is taken as the upper bound of the theoretically predicted pruning ratio. All results are shown in Figure 4.

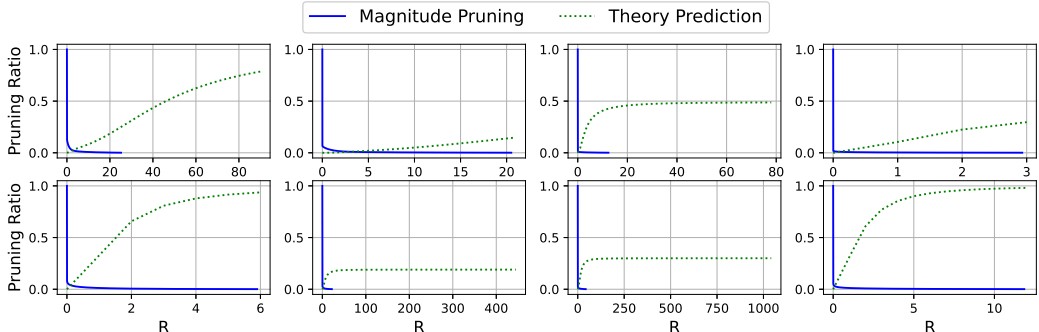

Figure 1: The theoretically predicted pruning ratio in eight tasks. The first row, from left to right, corresponds to FC5, FC12, AlexNet, and VGG16. The second row, from left to right, corresponds to ResNet18 and ResNet50 on CIFAR100, as well as ResNet18 and ResNet50 on TinyImagenet.

## B APPROXIMATE CALCULATION OF GAUSSIAN WIDTH

In practical experiments, determining the Gaussian width of the ellipsoid defined by the network loss function is a challenging task. There are two primary challenges encountered in this section: 1. the computation of eigenvalues for high-dimensional matrices poses significant difficulty; 2. the network fails to converge perfectly to the extremum, leading to a non-positive definite Hessian matrix for the loss function. In this section, we tackle these challenges through the utilization of a fast eigenspectrum estimation algorithm and an algorithm that approximates the Gaussian width of a deformed ellipsoid body. These approaches effectively address the aforementioned problems.

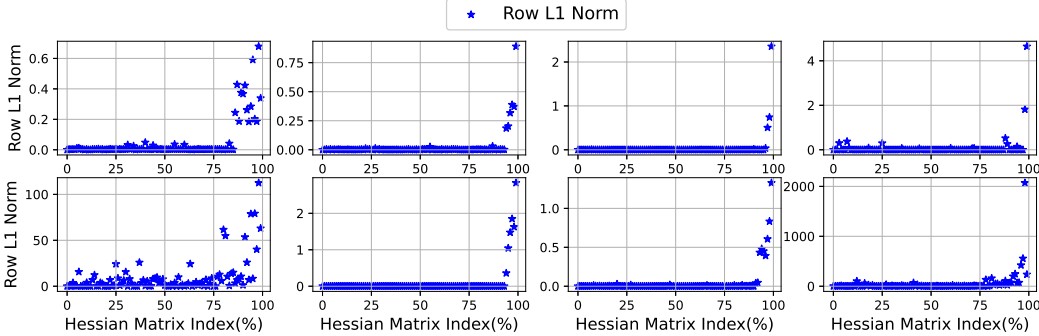

Figure 2: The statistical analysis of the L1 norm of the Hessian matrix in eight tasks. The first row, from left to right, corresponds to FC5, FC12, AlexNet, and VGG16. The second row, from left to right, corresponds to ResNet18 and ResNet50 on CIFAR100, as well as ResNet18 and ResNet50 on TinyImagenet.

## B.1 Improved SLQ (Stochastic Lanczos Quadrature) Spectrum Estimation

Calculating the eigenvalues of large matrices has long been a challenging problem in numerical analysis. One widely used method for efficiently computing these eigenvalues is the Lanczos algorithm, which is presented in Appendix B. However, due to the huge amount of parameters of the deep neural network, it is still impractical to use this method to calculate the eigenspectrum of the Hessian matrix of a deep neural network. To tackle this problem, Yao et al. (2020) proposed SLQ(Stochastic Lanczos Quadrature) Spectrum Estimation Algorithm, which estimates the overall eigenspectrum distribution based on a small number of eigenvalues obtained by Lanczos algorithm. This method enables the efficient computation of the full eigenvalues of large matrices. Algorithm 1 outlines the step-by-step procedure for the classic Lanczos algorithm, providing a comprehensive guide for its implementation. The algorithm requires the selection of the number of iterations, denoted as $m$, which determines the size of the resulting triangular matrix $\mathbf{T}$.

---

**Algorithm 1:** The Lanczos Algorithm

**Input:** a Hermitian matrix $\mathbf{A}$ of size $n \times n$, a number of iterations $m$
**Output:** a tridiagonal real symmetric matrix $\mathbf{T}$ of size $m \times m$
  initialization:
  1. Draw a random vector $\mathbf{v_1}$ of size $n \times 1$ from $\mathcal{N}(0,1)$ and normalize it;
  2. $\mathbf{w_1'} = \mathbf{A}\mathbf{v_1}$; $\alpha_1 = <\mathbf{w_1'}, \mathbf{v_1}>$; $\mathbf{w_1} = \mathbf{w_1'} - \alpha_1\mathbf{v_1}$;
  3.
  **for** $j = 2, ..., m$ **do**
    1). $\beta_j = \|\mathbf{w_{j-1}}\|$;
    2).
      **if** $\beta_j = 0$ **then**
        stop
      **else**
        $\mathbf{v_j} = \mathbf{w_{j-1}}/\beta_j$
      **end if**
    3). $\mathbf{w_j'} = \mathbf{A}\mathbf{v_j}$;
    4). $\alpha_j = <\mathbf{w_j'}, \mathbf{v_j}>$;
    5). $\mathbf{w_j} = \mathbf{w_j'} - \alpha_j\mathbf{v_j} - \beta_j\mathbf{v_{j-1}}$;
  **end for**
  4. $\mathbf{T}(i,i) = \alpha_i$, $i = 1, \ldots, m$;
    $\mathbf{T}(i, i+1) = \mathbf{T}(i+1, i) = \beta_i$, $i = 1, \ldots, m-1$.
  **return T**

---

In general, the Lanczos algorithm is not capable of accurately computing zero eigenvalues, and this limitation becomes more pronounced when the SLQ algorithm has a small number of iterations. Similarly, vanishingly small eigenvalues are also ignored by Lanczos. However, in a well-trained large-scale deep neural network, the experiment found that the network loss function hessian matrix has a large number of zero eigenvalues and vanishingly small eigenvalues. In the Gaussian width of the ellipsoid, the zero eigenvalues and vanishingly small eigenvalues have the same effect on the width (insensitive to other parameters), and we collectively refer to these eigenvalues as the "important" eigenvalues. We divide the weight into 100 parts from small to large, calculate the second-order derivative (including partial derivative) of smallest weight in each part, and sum the absolute values of all second-order derivatives of the weight, which corresponds to $l_1$-norm of a row in hessian matrix, and the row $l_1$-norm is zero or a vanishingly small corresponds to an "important" eigenvalue, the experimental results can be seen in the first column of Figure **??**, from which the number of missing eigenvalues of the SLQ algorithm can be estimated, we then add the same number of 1e-30 as the missing eigenvalues in the Hessian matrix eigenspectrum.SLQ algorithm parameters adjustment is discribed in Table 6 and the statistical analysis of the L1 norm of Hessian matrix rows for all experiments is presented in Figure 2. For details of the SLQ algorithm, see Algorithm 2

---

**Algorithm 2:** SLQ(Stochastic Lanczos Quadrature) Spectrum Estimation Algorithm

---

**Input:** A hermitian matrix $\mathbf{A}$ of size $n \times n$, Lanczos iterations $m$, ESD computation iterations $l$, gaussian kernel $f$ and variance $\sigma^2$.

**Output:** The spectral distribution of matrix $\mathbf{A}$

  **for** $i = 2, ..., l$ **do**

    1). Get the tridiagonal matrix $\mathbf{T}$ if size $m \times m$ through Lanczos algorithm;

    2). Compute $\tau_k^{(i)}$ and $\lambda_k^{(i)}$ from $\mathbf{T}$;

    3). $\phi_\sigma^i(t) = \sum_{k=1}^m \tau_k^{(i)} f(\lambda_k^{(i)}; t, \sigma)$;

  **end for**

  4). $\phi(t) = \frac{1}{l} \sum_{i=1}^l \phi_\sigma^i(t)$

  **return** $\phi(t)$

---

### B.2 GAUSSIAN WIDTH OF THE DEFORMED ELLIPSOID

After effective training, it is generally assumed that a deep neural network will converge to the global minimum of its loss function. However, in practice, even after meticulous tuning, the network tends to oscillate around the minimum instead of converging to it. This leads to that the Hessian matrix of the loss function would be non-positive definite, and the resulting geometric body defined by this matrix would change from an expected ellipsoid to a hyperboloid, which is unfortunately nonconvex. To quantify the Gaussian width of the ellipsoid corresponding to the perfect minima, we propose to approximate it by convexifying the deformed ellipsoid through replacing the associated negative eigenvalues with its absolute value. This processing turns out to be very effective, as demonstrated by the experimental results.

**Lemma 1** *Consider a well-trained neural network $M$ with weights $\mathbf{w}$, whose loss function defined by $\mathbf{w}$ has a Hessian matrix $\mathbf{H}$. Due to the non-positive definiteness of $\mathbf{H}$, there exist negative eigenvalues. Let the eigenvalue decomposition of $\mathbf{H}$ be $\mathbf{H} = \mathbf{v}^T \mathbf{\Sigma} \mathbf{v}$, where $\mathbf{\Sigma}$ is a diagonal matrix of eigenvalues. Let $\mathbf{D} = \mathbf{v}^T |\mathbf{\Sigma}| \mathbf{v}$, where $|\cdot|$ means absolute operation. the geometric objects defined by H and D are $S(\epsilon) := \{\mathbf{w} \in \mathbb{R}^D : \frac{1}{2}\mathbf{w}^T \mathbf{H} \mathbf{w} \leq \epsilon\}$ and $\hat{S}(\epsilon) := \{\mathbf{w} \in \mathbb{R}^D : \frac{1}{2}\mathbf{w}^T \mathbf{D} \mathbf{w} \leq \epsilon\}$, then:*

$$w(S(\epsilon)) \approx w(\hat{S}(\epsilon)) \tag{1}$$

The proof of Lemma 1 is in Appendix C.2. Lemma 1 indicates that if the deep neural network converges to a vicinity of the global minimum of the loss function, the Gaussian width of the deformed ellipsoid body can be approximated by taking the convex hull of $S(\epsilon)$. Experimental results demonstrate that the two approximation methods, namely setting negative features to zero and taking absolute values, yield nearly indistinguishable outcomes.

## C THEORETICAL PART SUPPLEMENT

In this section, we provide details regarding the threshold of network pruning ratio, specifically, the dimension of the sublevel sets of quadratic wells.

### C.1 GAUSSIAN WIDTH OF THE QUADRATIC WELL

Gaussian width is an extremely useful tool to measure the complexity of a convex body. In our proof, we will use the following expression for its definition:

$$w(S) = \frac{1}{2}\mathbb{E} \sup_{\mathbf{x}, \mathbf{y} \in S} \langle \mathbf{g}, \mathbf{x} - \mathbf{y} \rangle, \mathbf{g} \sim \mathcal{N}(\mathbf{0}, \mathbf{I}_{D \times D})$$

Concentration of measure is a universal phenomenon in high-dimensional probability. Basically, it says that a random variable which depends in a smooth way on many independent random variables (but not too much on any of them) is essentially *constant*.**?**

**Theorem C. 1** *(Gaussian concentration) Consider a random vector* $\mathbf{x} \sim \mathcal{N}(\mathbf{0}, \mathbf{I}_n)$ *and an L-Lipschitz function* $f : \mathbb{R}^n \to \mathbb{R}$ *(with respect to the Euclidean metric). Then for* $t \geq 0$

$$\mathbb{P}(|f(\mathbf{x}) - \mathbb{E}f(\mathbf{x})| \geq t) \leq \epsilon, \quad \epsilon = e^{-\frac{t^2}{2L^2}}.$$

Therefore, if $\epsilon$ is small, $f(\mathbf{x})$ can be approximated as $f(\mathbf{x}) \approx \mathbb{E}f(\mathbf{x}) + \sqrt{-2L^2 \mathrm{ln}\epsilon}$.

**Lemma C. 1** *Given a random vector* $\mathbf{x} \sim \mathcal{N}(\mathbf{0}, \mathbf{I}_n)$ *and the inverse of a positive definite Hessian matrix* $\mathbf{Q} = \mathbf{H}^{-1}$, *where* $\mathbf{H} \in \mathbb{R}^{n \times n}$, *we have:*

$$\mathbb{E}\sqrt{\mathbf{x}^\mathbf{T}\mathbf{Q}\mathbf{x}} \approx \sqrt{\mathbb{E}\mathbf{x}^\mathbf{T}\mathbf{Q}\mathbf{x}}$$

*Proof.*
1.) Concentration of $\mathbf{x}^\mathbf{T}\mathbf{Q}\mathbf{x}$

Define $f(\mathbf{x}) = \mathbf{x}^\mathbf{T}\mathbf{Q}\mathbf{x}$, we have

$$\begin{aligned}
f(\mathbf{x}) &= \mathbf{x}^\mathbf{T}\mathbf{Q}\mathbf{x} \\
&= \mathbf{x}^\mathbf{T}\mathbf{U}\mathbf{\Sigma}\mathbf{U}^\mathbf{T}\mathbf{x} && \text{Eigenvalue Decomposition of } \mathbf{Q}: \ \mathbf{Q} = \mathbf{U}\mathbf{\Sigma}\mathbf{U}^\mathbf{T}. \\
&= \sum_{i=1}^{n} \lambda_i x_i^2 && \text{Invariance of Gaussian under rotation.}
\end{aligned}$$

where $\lambda_i$ is the eigenvalue of $\mathbf{Q}$. The lipschitz constant $L_f$ of function $f(\mathbf{x})$ is :

$$L_f = \max(|\frac{\partial f}{\partial \mathbf{x}}|) = \max(|2\lambda_i x_i|)$$

Let $g(x_i) = 2\lambda_i x_i$, whose lipschitz constant is $L_g = |2\lambda_i|$. Invoking Theorem 1, we have:

$$\begin{aligned}
g(x_i) &\approx \mathbb{E}g(x_i) + \sqrt{-2(2\lambda_i)^2 \mathrm{ln}\epsilon_1} \\
&= \sqrt{-8\lambda_i^2 \mathrm{ln}\epsilon_1}.
\end{aligned}$$

Therefore, the lipschitz constant of $f(\mathbf{x})$ can be approximated by:

$$L_f = max(\sqrt{-8\lambda_i^2 \mathrm{ln}\epsilon_1}) \quad = \sqrt{-8\mathrm{ln}\epsilon_1}\lambda_{max}$$

Invoking Theorem 1 again, we establish the concentration of $f(\mathbf{x})$ as follows:

$$\begin{aligned}
f(\mathbf{x}) &\approx \mathbb{E}f(\mathbf{x}) + \sqrt{-2(L_f)^2 \mathrm{ln}\epsilon_2} && \text{Theorem 1.} \\
&= \mathbb{E}f(\mathbf{x}) + 4\sqrt{\mathrm{ln}\epsilon_1 \mathrm{ln}\epsilon_2}\lambda_{max}
\end{aligned}$$

2.) Jensen ratio of $\sqrt{\mathbf{x}^\mathbf{T}\mathbf{Q}\mathbf{x}}$:

$$\begin{aligned}
\mathbb{E}\sqrt{f(\mathbf{x})} &\approx \mathbb{E}\sqrt{\mathbb{E}f(\mathbf{x}) + 4\sqrt{\mathrm{ln}\epsilon_1 \mathrm{ln}\epsilon_2}\lambda_{max}} && \text{Concentration of } f(\mathbf{x}). \\
&\approx \sqrt{\mathbb{E}f(\mathbf{x})} + \frac{2\sqrt{\mathrm{ln}\epsilon_1 \mathrm{ln}\epsilon_2}\lambda_{max}}{\sqrt{\mathbb{E}f(\mathbf{x})}} && \text{Taylor Expansion.}
\end{aligned}$$

Therefore, the Jensen ratio of $\sqrt{f(\mathbf{x})}$ equals:

$$\begin{aligned}
\frac{\mathbb{E}\sqrt{f(\mathbf{x})}}{\sqrt{\mathbb{E}f(\mathbf{x})}} &= 1 + 2\sqrt{\mathrm{ln}\epsilon_1 \mathrm{ln}\epsilon_2}\frac{\lambda_{max}}{\sum_{i=1}^{n} \lambda_i} \\
&= 1 + \delta
\end{aligned}$$

If $\mathbf{Q}$ is a Wishart matrix, i.e., $\mathbf{Q} = \mathbf{A}^T\mathbf{A}$, where $\mathbf{A}$ is a random matrix whose elements are independently and identically distributed with unit variance, according to the Marchenko-Pastur law (Tao,

2012), the maximum eigenvalue of $\mathbf{Q}$ is approximately $4n$ and the trace of $\mathbf{Q}$ is approximately $n^2$. Therefore, the above Jensen ratio approaches to 1 with decaying rate $\mathcal{O}(\frac{1}{n})$.

For the inverse of a positive definite Hessian matrix which is of our concern, we take $\epsilon_1 = \epsilon_2 = 10^{-4}$, numerical simulations show that when the dimension $n = 10^5$, the corresponding $\delta$ in the above Jensen ratio is on the order of $10^{-3}$, which is in good agreement with the theoretical value and is arguably negligible. Similar as the case of the above-discussed Wishart matrix, when the dimension $n$ increases, the value of $\delta$ will further decrease.

**Definition C. 1** *(Definition of ball) A (closed) ball $B(c,r)$ (in $\mathbb{R}^D$) centered at $c \in \mathbb{R}^D$ with radius $r$ is the set*

$$B(c,r) := \{\mathbf{x} \in \mathbb{R}^D : \mathbf{x}^T\mathbf{x} \le r^2\}$$

*The set $B(0,1)$ is called the $unit$ $ball$. An ellipsoid is just an affine transformation of a ball.*

**Lemma C. 2** *(Definition of ellipsoid). An $ellipsoid$ $S$ centered at the origin is the image $L(B(0,1))$ of the unit ball under an $invertible$ linear transformation $L : \mathbb{R}^D \to \mathbb{R}^D$. An ellipsoid centered at a general point $c \in \mathbb{R}^D$ is just the translate $c + S$ of some ellipsoid $S$ centered at the origin.*

*Proof.*

$$
\begin{aligned}
L(B(0,1)) &= \{\mathbf{Lx} : \mathbf{x} \in B(0,1)\} \\
&= \{\mathbf{y} : \mathbf{L}^{-1}\mathbf{y} \in B(0,1)\} \\
&= \{\mathbf{y} : (\mathbf{L}^{-1}\mathbf{y})^T\mathbf{L}^{-1}\mathbf{y} \le 1\} \\
&= \{\mathbf{y} : \mathbf{y}^T(\mathbf{LL}^T)^{-1}\mathbf{y} \le 1\} \\
&= \{\mathbf{y} : \mathbf{y}^T\mathbf{Q}^{-1}\mathbf{y} \le 1\}
\end{aligned}
$$

where $\mathbf{Q} = \mathbf{LL}^T$ is **positive definite**.
The radius $r_i$ along principal axis $\mathbf{e}_i$ obeys $r_i^2 = \frac{1}{\lambda_i}$, where $\lambda_i$ is the eigenvalue of $\mathbf{Q}^{-1}$ and $\mathbf{e}_i$ is eigen vector.

**Lemma C. 3** *(Gaussian width of ellipsoid). Let $S$ be an ellipsoid in $\mathbb{R}^D$ defined by the positive definite matrix $\mathbf{H} \in \mathbb{R}^{D \times D}$:*

$$S(\epsilon) := \{\mathbf{w} \in \mathbb{R}^D : \frac{1}{2}\mathbf{w}^T\mathbf{Hw} \le \epsilon\}$$

*Then $w(S)^2$ or the Gaussian width squared of the ellipsoid satisfies:*

$$w(S)^2 \approx 2\epsilon\mathrm{Tr}(\mathbf{H}^{-1}) = \sum_i r_i^2$$

*where $r_i = \sqrt{2\epsilon/\lambda_i}$ with $\lambda_i$ is $i$-th eigenvalue of $\mathbf{H}$.*

*Proof.* Let $\mathbf{g} \sim \mathcal{N}(\mathbf{0}, \mathbf{I}_{D \times D})$ and $\mathbf{LL}^T = 2\epsilon\mathbf{H}^{-1}$. Then:

$$
\begin{aligned}
w(L(B_2^n)) &= \frac{1}{2}\mathbb{E}\, sup_{\mathbf{x},\mathbf{y}\in B(0,1)} <\mathbf{g}, \mathbf{Lx} - \mathbf{Ly}> \\
&= \frac{1}{2}\mathbb{E}\, sup_{\mathbf{x},\mathbf{y}\in B(0,1)} <\mathbf{gL}, \mathbf{x} - \mathbf{y}> \\
&= \mathbb{E}\|\mathbf{gL}\|_2 &&\text{Definition of Ball.} \\
&= \mathbb{E}\sqrt{(\mathbf{gLL^Tg^T})} &&\|\mathbf{g}\|_2 = \sqrt{\mathbf{gg^T}},\text{ where } \mathbf{g} \in \mathbb{R}^{1 \times D}. \\
&= \mathbb{E}\sqrt{2\epsilon\mathbf{gH^{-1}g^T}} \\
&\approx \sqrt{2\epsilon\mathbb{E}[\mathbf{gH^{-1}g^T}]} &&\text{Lemma 1.} \\
&= \sqrt{2\epsilon\mathrm{Tr}(\mathbf{H}^{-1})} &&\text{Invariance of Gaussian under rotation.}
\end{aligned}
$$

Thus, $w(S)^2 \approx 2\epsilon\mathrm{Tr}(\mathbf{H}^{-1}) = \sum_i r_i^2$.

C.2  GAUSSIAN WIDTH OF THE DEFORMED ELLIPSOID

Generally, it is assumed that the gradient descent algorithm will converge to a minimum point. However, in practice, even with small learning rates, the network may oscillate near the minimum point and not directly converge to it, but rather get very close to it. As a result, the actual Hessian matrix is often not positive definite and its eigenvalues may have negative values.

**Lemma C. 4** *Let the Hessian matrix at the minimum point be denoted by $\mathbf{H}$ with eigenvalue $\lambda_i$, and the Hessian matrix at an oscillation point be denoted by $\hat{\mathbf{H}}$ with eigenvalue $\hat{\lambda}_i$. The negative eigenvalues of $\hat{\mathbf{H}}$ have small magnitudes.*

*Proof.* Let the weights at the minimum point be denoted by $\mathbf{w}_0$ and the Hessian matrix at an oscillation point be denoted by $\hat{\mathbf{w}}_0$. Consider a loss function $L$ and a loss landscape defined by $L(\mathbf{w})$, taking taylor expansion of $L(\mathbf{w})$ at $\mathbf{w}_0$:

$$L(\mathbf{w}) = L(\mathbf{w}_0) + \frac{1}{2}(\mathbf{w} - \mathbf{w}_0)^T \mathbf{H}(\mathbf{w} - \mathbf{w}_0) + R(\mathbf{w}_0)$$

Let $\hat{\mathbf{w}}_0 = \mathbf{w}_0 + \mathbf{v}$ with $\mathbf{v}$ is closed to $\mathbf{0}$:

$$L(\hat{\mathbf{w}}_0) = L(\mathbf{w}_0 + \mathbf{v})$$
$$= L(\mathbf{w}_0) + \frac{1}{2}\mathbf{v}^T \mathbf{H}\mathbf{v} + R(\mathbf{w}_0 + \mathbf{v})$$

Therefore, the second order derivative of $L(\hat{\mathbf{w}}_0)$ is:

$$L^{''}(\mathbf{w}) = L^{''}(\mathbf{w}_0 + \mathbf{v})$$
$$= \mathbf{H} + R^{''}(\mathbf{w}_0 + \mathbf{v})$$
$$\approx \mathbf{H}$$

where $L^{''}(\mathbf{w}) = \hat{\mathbf{H}}$, Let $\mathbf{H} = \hat{\mathbf{H}} + \mathbf{H}_0$ with $\mathbf{H}_0$ is closed to $\mathbf{0}$, considering the Weyl inequality:

$$\lambda_i(\mathbf{H}) - \hat{\lambda}_i(\hat{\mathbf{H}}) \leq \|\mathbf{H}_0\|_2$$

where $\|\mathbf{H}_0\|_2$ is small enough. So if $\hat{\lambda}_i(\hat{\mathbf{H}})$ is less than 0, since $\hat{\lambda}_i(\hat{\mathbf{H}}) \geq \lambda_i(\mathbf{H}) - \|\mathbf{H}_0\|_2$, its absolute value $|\hat{\lambda}_i(\hat{\mathbf{H}})| \leq \|\mathbf{H}_0\|_2 - \lambda_i(\mathbf{H}) \leq \|\mathbf{H}_0\|_2$, which means that the negative eigenvalues of the Hessian matrix have small magnitudes.

**Lemma C. 5** *For a sublevel set $S(\epsilon) := \{\mathbf{w} : \mathbf{w}^T \mathbf{H}\mathbf{w} \leq \epsilon\}$ defined by a matrix $\mathbf{H}$ with small magnitude negative eigenvalues. The Gaussian width of $S(\epsilon)$ can be estimated by obtaining the absolute values of the eigenvalues of the matrix $\mathbf{H}$.*

*Proof.* Assuming that the eigenvalue decomposition of $\mathbf{H}$ is $\mathbf{H} = \mathbf{v}^T \boldsymbol{\Sigma} \mathbf{v}$, where $\boldsymbol{\Sigma}$ is a diagonal matrix consisting of the eigenvalues of $\mathbf{H}$, let $\mathbf{D} = \mathbf{v}^T |\boldsymbol{\Sigma}| \mathbf{v}$ be a positive definite matrix and $\mathbf{M} = \mathbf{H} - \mathbf{D} = \mathbf{v}^T(\boldsymbol{\Sigma} - |\boldsymbol{\Sigma}|)\mathbf{v}$ be a negative definite matrix. Consider the definition of $S(\epsilon)$:

$$\mathbf{w}^T \mathbf{H}\mathbf{w} = \mathbf{w}^T(\mathbf{H} - \mathbf{D} + \mathbf{D})\mathbf{w}$$
$$= \mathbf{w}^T \mathbf{M}\mathbf{w} + \mathbf{w}^T \mathbf{D}\mathbf{w}$$
$$\leq \epsilon$$

Therefore, $S(\epsilon)$ can be expressed as $\mathbf{w}^T \mathbf{D}\mathbf{w} \leq \epsilon - \mathbf{w}^T \mathbf{M}\mathbf{w}$. Since the magnitudes of the negative eigenvalues of $\mathbf{H}$ are very small, we can assume that $\mathbf{w}^T \mathbf{M}\mathbf{w}$ is also small, and thus $\mathbf{w}^T \mathbf{D}\mathbf{w} \leq \epsilon - \mathbf{w}^T \mathbf{M}\mathbf{w}$ can be approximately equal to $\mathbf{w}^T \mathbf{D}\mathbf{w} \leq \epsilon$. As a result, we can estimate the Gaussian width of $S(\epsilon)$ by approximating it with the absolute values of the eigenvalues of $\mathbf{H}$.

**Corollary C. 1** *Consider a well-trained neural network $M$ with weights $\mathbf{w}$, whose loss function defined by $\mathbf{w}$ has a Hessian matrix $\mathbf{H}$. Due to the non-positive definiteness of $\mathbf{H}$, there exist negative eigenvalues. Let the eigenvalue decomposition of $\mathbf{H}$ be $\mathbf{H} = \mathbf{v}^T \boldsymbol{\Sigma} \mathbf{v}$, where $\boldsymbol{\Sigma}$ is a diagonal matrix of eigenvalues. Let $\mathbf{D} = \mathbf{v}^T |\boldsymbol{\Sigma}| \mathbf{v}$, where $|\cdot|$ means absolute operation. the geometric objects defined by $H$ and $D$ are $S(\epsilon) := \{\mathbf{w} \in \mathbb{R}^D : \frac{1}{2}\mathbf{w}^T \mathbf{H}\mathbf{w} \leq \epsilon\}$ and $\hat{S}(\epsilon) := \{\mathbf{w} \in \mathbb{R}^D : \frac{1}{2}\mathbf{w}^T \mathbf{D}\mathbf{w} \leq \epsilon\}$, then:*

$$w(S(\epsilon)) \approx w(\hat{S}(\epsilon))$$

# D ANALYSIS OF THE RELATIONSHIP BETWEEN THE UPPER AND LOWER BOUND

This section provided the proofs of the lower bound derivation and roughly analyzed how the lower bound changes when the upper bound varies.

## D.1 $D - m$ DIMENSION SUBLEVEL SET IS STILL AN ELLIPSOID

In the derivation of the lower bound for the pruning ratio threshold, we employed a $D - m$ dimensional loss sublevel set:

$$S(\mathbf{w}^2) = \{\mathbf{w}^2 \in \mathbb{R}^{D-m} : \mathcal{L}(\mathbf{w}^1, \mathbf{w}^2) \leq \epsilon\} \tag{2}$$

Perform Taylor expansion to $\mathcal{L}(\mathbf{w}^1, \mathbf{w}^2)$ with respect to $\mathbf{w}^2$, the sublevel set is represented as:

$$S(\mathbf{w}^2) = \{\mathbf{w}^2 \in \mathbb{R}^{D-m} : \frac{1}{2}(\mathbf{w}^2)^T \mathbf{H}^2 \mathbf{w}^2 \leq \epsilon\} \tag{3}$$

where $\mathbf{H}^2$ is the Hessian matrix of $\mathcal{L}(\mathbf{w}^1, \mathbf{w}^2)$ with respect to $\mathbf{w}^2$.

Given that the sublevel set $S(\epsilon, \mathbf{w}^*) = \{\hat{\mathbf{w}} \in \mathbb{R}^D : \frac{1}{2}\hat{\mathbf{w}}^T \mathbf{H}\hat{\mathbf{w}} \leq \hat{\epsilon}\}$ is an ellipsoid body, which implies that $\mathbf{H}$ is a positive definite matrix, it is evident that $\mathbf{H}^2$ is the principal submatrix of $\mathbf{H}$. Consequently, $\mathbf{H}^2$ is also a positive definite matrix, which implies that the sublevel set remains an ellipsoid.

## D.2 RELATIONSHIP BETWEEN THE UPPER AND LOWER BOUND

**Theorem 1 (Eigenvalue Interlacing Theorem)** *Suppose $\mathbf{A} \in \mathbb{R}^{n \times n}$ is symmetric, Let $\mathbf{B} \in \mathbb{R}^{m \times m}$ with $m < n$ be a principal submatrix(obtained by deleting both $i$-th row and $i$-th column for some values of $i$). Suppose $\mathbf{A}$ has eigenvalues $\lambda_1 \leq \cdots \leq \lambda_n$ and $\mathbf{B}$ has eigenvalues $\beta_1 \leq \cdots \leq \beta_m$. Then*

$$\lambda_k \leq \beta_k \leq \lambda_{k+n-m} \quad for \quad k = 1, \ldots, m \tag{4}$$

*And if $m = n - 1$,*

$$\lambda_1 \leq \beta_1 \leq \lambda_2 \leq \beta_2 \leq \cdots \leq \beta_{n-1} \leq \lambda_n \tag{5}$$

Theorem 1 indicates that: if $\mathbf{H}^2$ the principal submatrix of $\mathbf{H}$, where $\mathbf{H}$ is a positive definite matrix, when the eigenvalues of $\mathbf{H}$ increase, the eigenvalues of $\mathbf{H}^2$ also increase. Conversely, as the eigenvalues of $\mathbf{H}$ decrease, the eigenvalues of $\mathbf{H}^2$ decrease. In essence, this implies that the changes in the upper and lower bounds resulting from the eigenvalues occur in the same direction. The changes in the lower bound also exhibit the same direction as the variations in the upper bound of the pruning ratio threshold when the magnitude of network parameters increases or decreases. Investigating the exact numerical relationship between the upper and lower bound is left to future work.

# E FULL RESULTS

Here we present the full set of experiments performed for the results in the main text.

## E.1 SMALL WEIGHTS BENEFITS PRUNING

We verify that high flatness is not equal to high sparsity through hypothetical experiments. Considering that the hessian matrix of network $A$ and network $B_1, B_2, B_3, B_4$ share eigenvalues $\{\lambda_1, \lambda_2, \ldots, \lambda_n\}$, the weight magnitude of network $B_1, B_2, B_3, B_4$ is 2,3,4,5 times that of network $A$, we take the eigenvalues and weights from a FC network trained without regularization. In this way, the gap between the curves will be more obvious. For other networks, the trend of the curve gap is consistent, the prediction of the network pruning ratio is shown in the Figure. 3. It is observed from Figure. 3 that as the magnitude of network weights increases, the capacity of the network to tolerate pruning decreases. The pruning ratio threshold is affected not only by loss flatness but also the magnitude of weights. This finding, on the other hand, provides further evidence of the effectiveness of the $l_1$-norm in pruning tasks.

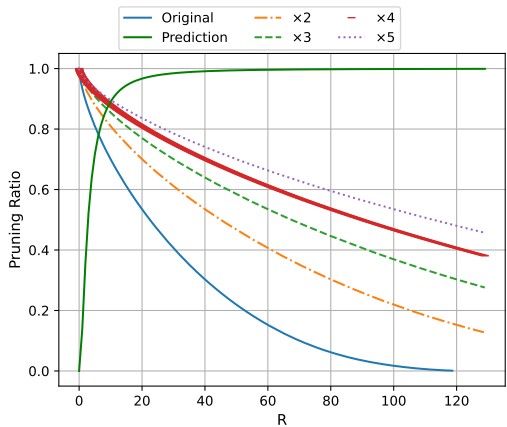

Figure 3: Pruning ratio prediction on different weight magnitude.

## E.2 STATISTICAL INFORMATION OF WEIGHTS IN VARIOUS EXPERIMENTS

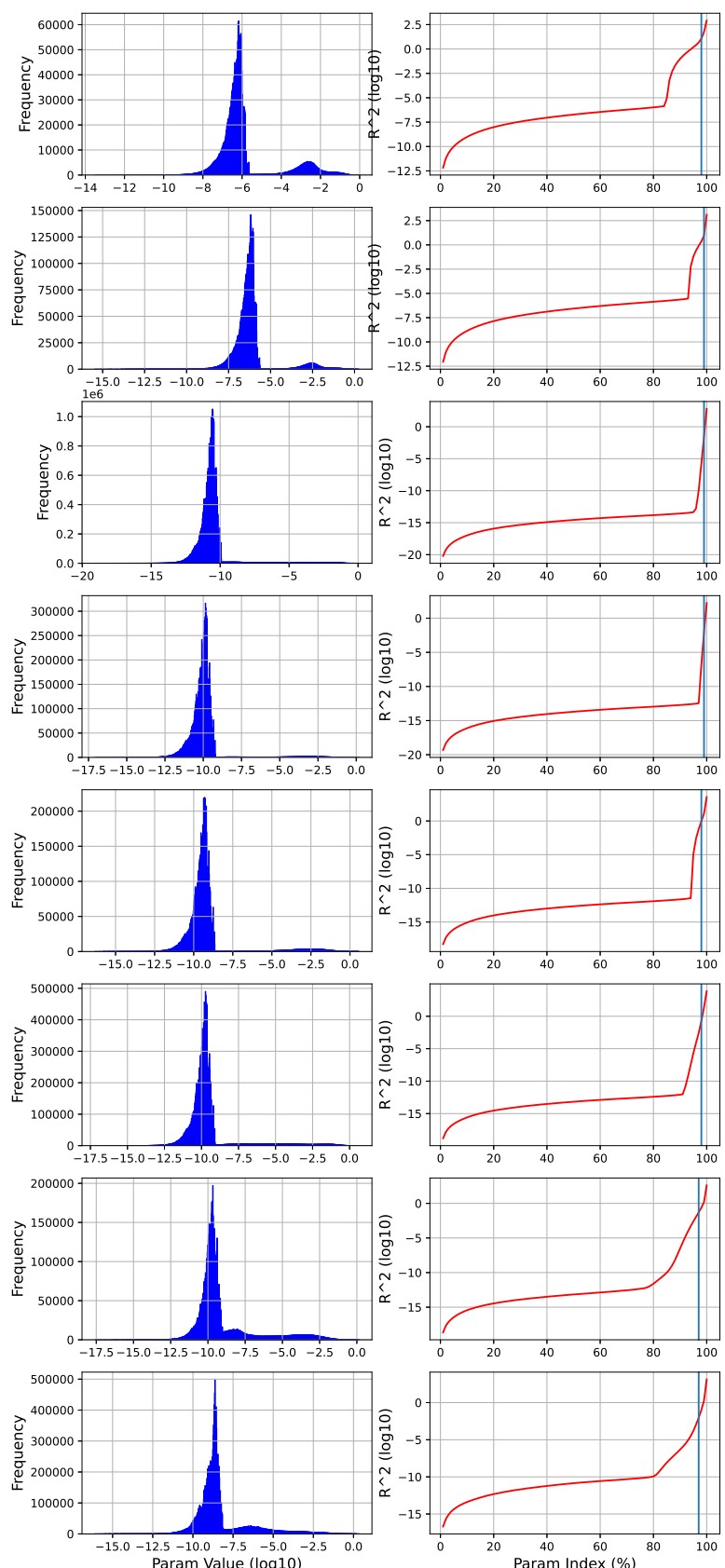

Figure 4: The same plots as Fig. 2(b) and Fig. 2(c) on CIFAR10 FC5,FC12,AlexNet,VGG16, and CIFAR100 ResNet18,ResNet50 and TinyImageNet ResNet18,ResNet50.