# OpenReview forum: "How Sparse Can We Prune A Deep Network: A Geometric Viewpoint"
_ICLR.cc/2024/Conference — Submitted to ICLR 2024_

### Official Review · Reviewer_Br8B · 2023-10-30

**Soundness:** 3 good
**Presentation:** 3 good
**Contribution:** 3 good
**Rating:** 6
**Confidence:** 4

**Summary:**

This paper proposes the question that how sparse can we prune the neural network without sacrifice the performance and attempt to use Approximate Kinematics Formula and statistical dimension to find the limit of pruning. In detail, the authors attempt to find the lower and upper bound of the limit sparsity and use both theoretical and experimental results to illustrate that the lower and upper bound are close. In addition, the influence of the loss function and the magnitude of weights are discussed based on the principle proposed by the authors. Many phenomena accompanied with existing pruning such as magnitude pruning is better than random pruning and iterative pruning is better than one shot pruning are also explained with the proposed principle.

**Strengths:**

The perspective of this paper is very interesting and helpful. Firstly the authors alter the original theorem from[1] to show that the dimension of the loss sub level set and the sparse level should fill the whole space to ensure a pruning without sacrificing the performance. It is in accordance with intuition. Then the estimation of the sub level set is transformed into the form of quadratic form of the hessian matrix which is then approximated by the term including the eigen value. In this way the authors transform the original problem into the study of eigen value of the hessian matrix. It makes the problem much clear.
The limit derived by induction is also verified by extensive experiments.
In addition, the authors also use several concept plots to illustrate their key idea.

**Weaknesses:**

For Table 3, I am confused by the margin of improvement. Could the author give a detailed illustration of your GOP. From my point of view, you use the magnitude pruning and the magnitude is determined by l1 norm. In addition you prune the network globally. However, for pruning using magnitude, using l1 or l2 norm will not alter the rank of these weights which means the remaining weights after pruning is the same. Then why your GOP can improve significantly against LTH. In addition, could the authors further validate the theory on large scale dataset such ImageNet or Place365?

For LTH, the key point is to keep the same initialization when doing  training. Could the theory provide an explanation for this phenomenon?

For the upper bound and lower bound part, the lower bound means the pruning limit of the network. However, I am confused by the upper bound here. It seems that the authors mean to illustrate firstly the limit we can not find a sparser one without sacrificing the performance, then for some sparsity we can find one sparse network satisfy the requirement. I am not sure whether call it upper bound is suitable.

For the part of Approximate Kinematics Formula from [1], a concrete introduction could be given in the paper for better understanding, for this part is the key component of this paper.


[1] Dennis Amelunxen, Martin Lotz, Michael B. McCoy, and Joel A. Tropp. Living on the edge: Phase transitions in convex programs with random data, 2014.

**Questions:**

Please refer to Weakness.

---

> ### Author Response · Authors · 2023-11-23
> **Response to Reviewer Br8B**
>
> Thanks for your positive comments! The concerns and questions in the review as addressed as follows.
>
> **Question 1: For Table 3, I am confused by the margin of improvement. Could the author give a detailed illustration of your GOP. From my point of view, you use the magnitude pruning and the magnitude is determined by l1 norm. In addition you prune the network globally. However, for pruning using magnitude, using l1 or l2 norm will not alter the rank of these weights which means the remaining weights after pruning is the same. Then why your GOP can improve significantly against LTH. In addition, could the authors further validate the theory on large scale dataset such ImageNet or Place365?**
>
> A brief description of the GOP algorithm is provided in Section 2 of our paper. Basically, GOP runs as follows: we modify the original loss function with an $l_1$ regularization term and then perform network training, whose final outcome of weight will be pruned based on the magnitude.
>
> As for the mentioned $l_1$ norm against $l_2$ norm, indeed they make no difference when conducting the magnitude-based pruning. The key enabler of our algorithm as compared with the LTH lies in that the pruning ratio in GOP is determined by formulas, in a global and one-shot way, while the pruning ratio in LTH is iterative and predetermined in nature, easily causing performance loss due to that important parameters might be removed inappropriately during the pruning iterations.
>
> As for the large-scale dataset, due to the limited computational resources in our lab, experiments based on the ImageNet dataset is of difficulty for now. Therefore, we chose to use tiny version of Imagenet for validation, which shows that our theory aligns well with experimental outcomes. Relevant results can be found in Section 5. If this paper was finally accepted and experiments were finished then, we will provide the  validation results on Imagenet  in the final version.
>
> **Question 2: For LTH, the key point is to keep the same initialization when doing training. Could the theory provide an explanation for this phenomenon?**
>
> Since our theory is based on global and one-shot pruning, though  it can achive better  performance than the multi-shot pruing, say LTH, currently it cannot provide explanations for phenomena in the latter. We do think this is a much worthy-to-explore question.
>
> **Question 3: For the upper bound and lower bound part, the lower bound means the pruning limit of the network. However, I am confused by the upper bound here. It seems that the authors mean to illustrate firstly the limit we can not find a sparser one without sacrificing the performance, then for some sparsity we can find one sparse network satisfy the requirement. I am not sure whether call it upper bound is suitable.**
>
> By upper bound we mean that  we can find a weight of given sparsity without sacrificing performance with probability 1. In other words, the fundamental limit of the sparsity is smaller than the value given by the upper bound.

---

> ### Author Response · Authors · 2023-11-23
> **Response to Reviewer Br8B**
>
> **Question 4: For the part of Approximate Kinematics Formula from [1], a concrete introduction could be given in the paper for better understanding, for this part is the key component of this paper.**
>
> Thanks for the suggestion. We have modified the main idea description in a more intuitive manner in the revised version. Specifically, we have provided an intuitive description of the key tool in our work, i.e., the Approximate Kinematics Formula. Please refer to the following two excerpts:
>
> In Section 1:
>
> > Despite the above progress, it however still remains elusive about the *fundamental limit* of network pruning while maintaining the performance. To tackle this problem, we'll take a first principle approach by imposing the sparsity constraint directly on the loss function, thus converting the original pruning limit problem to a set intersection problem, i.e., deciding whether the *$k$-sparse set* intersects with the *loss sublevel set* (i.e., the set of weights whose corresponding loss is no larger than the original loss plus  tolerance $\epsilon$)  and obtaining the smallest value of $k$.
>
> >
> > Intuitively speaking, the larger the loss sublevel set (higher complexity), the smaller the sparse set required for intersection, i.e., the network can be more sparse. To rigorously characterize the complexity of a set, by exploiting the high dimensional nature of DNNs, we are able to harness the notion of *statistical dimension* and *Gaussian width* in high dimensional convex geometry. Through this geometric perspective, it's possible for us to take advantage of the universal *concentration effect* [1] in the high-dimensional world, so as to get sharp results about the above set intersection problem. In specific, we will exploit the powerful Approximate Kinematics Formula in high-dimensional geometry, which roughly says that for two convex cones, if the sum of their statistical dimension exceeds the ambient dimension, then these two cones would intersect with probability 1, otherwise they would intersect with probability 0. We notice that a sharp phase transition emerge here, thus enabling a precise and succinct characterization of the fundamental limit of network pruning.
>
> and in Section 2:
>
> > To characterize the sufficient and necessary condition of the set (or cone) intersection, the Approximate kinematics Formula is a powerful and sharp result, which basically says that for   two convex cones (or generally, sets), if the sum of their statistical dimension exceeds the ambient dimension, then these two cones would intersect with probability 1, otherwise they would intersect with probability 0.
>
>  [1] Basically, the concentration effect says that a function of a *large amount* of independent (or weakly dependent) variables tends to concentrate to its expectation value. Notable examples include the Johnson-Lindenstrauss lemma and related results in Compressive Sensing.

---

### Official Review · Reviewer_yqv3 · 2023-10-30

**Soundness:** 3 good
**Presentation:** 3 good
**Contribution:** 2 fair
**Rating:** 5
**Confidence:** 4

**Summary:**

The paper proposed to directly enforce the sparsity constraint on the original loss function to find a s parse sub-network. It also introduced methods to find the pruning rate by using 'Network Pruning Approximate Kinematic' which is driven from Approximate Kinematics Formula in high-dimensional geometry.

**Strengths:**

1. It provides theoretical justification of the pruning rates, i.e. lower bound or upper bound of the pruning rates.
2. It utilizes several tools for improving the computational efficiency when handling the Hessian matrix like employing Hessian matrix sampling.
3. Experimental results on several tasks and settings are provided.

**Weaknesses:**

1. In Eq.3, the proposed method uses task loss + $L_1$ regularization as the pruning objective. In Eq.13 of section 3.3, the authors considered a well-trained deep neural network with weights $w^*$. The setting in Eq.13 seems not related to the pruning objective considered in Eq.3. If this is the case, then the theoretical justification of the pruning rates is only for a deep neural network trained normally instead of trained under the pruning objective given in Eq.3.
2. Following the first point, the pruning objective provided in Eq.3 introduces a hyperparameter $\lambda$, which controls the regularization strength, and thus the sparsity of the model. A very straightforward way to determine the pruning rate under this setting is to count number of zeros in the original weight matrix, which will be much more efficient than using Hessian matrix to calculate the theoretical pruning rate. I understand that the theoretical pruning rate may also make values with small magnitude to be pruned, however, you can achieve the similar results by simply increasing $\lambda$ in Eq.3. If you think $L_1$ regularization cannot accurately control the pruning rate, then $L_0$ regularization [1] can achieve this, which is also differentiable.  As a result, I doubt the practical usage of the proposed method.
3. Since the theoretical analysis did not include $\lambda$ in Eq.3, it brings some additional problems. For example, modern deep neural network training could be very expensive, assume we trained a model, and we calculate the theoretical pruning rate. But this pruning rate does not meet our expectation, then the only thing we can do is to adjust the $\lambda$ and retrain the model. We never know the theoretical pruning rate before the model is fully trained. As a result, the theoretical pruning rate does not give a better guidance when we want to train a model given an expected pruning rate.
4. The pruning method itself is not novel, it is simply $L_1$ regularized training + magnitude pruning.

[1] LEARNING SPARSE NEURAL NETWORKS THROUGH L0 REGULARIZATION. https://arxiv.org/pdf/1712.01312.pdf

**Questions:**

1. Can the proposed method be used for dense models to predict its pruning rates? If it can be used, could you provide some results?
2. Do you try your methods on large-scale datasets like ImageNet? Does the computation of the Hessian Matrix become a bottleneck for large-scale datasets?
3. This is an open question just for discussion. Is it possible for your current theoretical framework to incorporate $\lambda$ to predict the final theoretical pruning rate? Is it possible for your current framework to use the theoretical pruning rate of an early trained model to predict the pruning rate of the fully trained model?

---

> ### Author Response · Authors · 2023-11-23
> **Response to Reviewer yqv3**
>
> We appreciate much for the reviewer's insightful and detailed feedback.
>
> ***Weaknesses 1: In Eq.3, the proposed method uses task loss + $l_1$ regularization as the pruning objective. In Eq.13 of section 3.3, the authors considered a well-trained deep neural network with weights ${\bf w}^*$. The setting in Eq.13 seems not related to the pruning objective considered in Eq.3. If this is the case, then the theoretical justification of the pruning rates is only for a deep neural network trained normally instead of trained under the pruning objective given in Eq.3.**
>
> Good question. The short answer is: Eq. 13 is for the purpose of theoretical analysis (of fundamental pruning limit), while Eq. 3 (Eq. 2 in the revised revision) is for the consideration of computational complexity. Specifically, Eq. 13 describes the loss sublevel set of the original loss function, which is of critical importance to the pruning limit of the original network. On the other hand, to solve the optimization problem of pruning limit, i.e., Eq. 1 (which directly corresponds to Eq. 13) , we have to relax the involved non-convex $l_0$ norm to $l_1$ norm, thus giving rise to Eq. 3. Surprisingly, the experimental result of pruning limit by running Eq. (3) is almost identical to the theoretical result originating from Eq. (13). The deeper connection between them is currently under our investigation.
>
> **Weaknesses 2: Following the first point, the pruning objective provided in Eq.3 introduces a hyperparameter $\lambda$, which controls the regularization strength, and thus the sparsity of the model. A very straightforward way to determine the pruning rate under this setting is to count number of zeros in the original weight matrix, which will be much more efficient than using Hessian matrix to calculate the theoretical pruning rate. I understand that the theoretical pruning rate may also make values with small magnitude to be pruned, however, you can achieve the similar results by simply increasing $\lambda$ in Eq.3. If you think $l_1$ regularization cannot accurately control the pruning rate, then $l_0$ regularization [1] can achieve this, which is also differentiable. As a result, I doubt the practical usage of the proposed method.**
>
> Due to space constraints, we did not include the comparison between $l_0$ based pruning and $l_1$ based pruning. The pruning algorithm based on $l_1$ regularization does not directly set the model parameters to zero, rather  a pruning procedure is needed after training. Therefore, counting the number of zeros in the weight matrix cannot be used to determine sparsity directly, as the network remains a dense model.
>
> We acknowledge that the algorithm presented in [1] might be more efficient than selecting $\lambda$ for $l_1$ based pruning algorithm. However, this is not the primary focus of our paper, and designing an algorithm for selecting $\lambda$ is a much valuable direction, which is exactly our ongoing research.
>
> [1] LEARNING SPARSE NEURAL NETWORKS THROUGH L0 REGULARIZATION. https://arxiv.org/pdf/1712.01312.pdf
>
> **Weaknesses 3: Since the theoretical analysis did not include $\lambda$ in Eq.3, it brings some additional problems. For example, modern deep neural network training could be very expensive, assume we trained a model, and we calculate the theoretical pruning rate. But this pruning rate does not meet our expectation, then the only thing we can do is to adjust the $\lambda$ and retrain the model. We never know the theoretical pruning rate before the model is fully trained. As a result, the theoretical pruning rate does not give a better guidance when we want to train a model given an expected pruning rate.**
>
> Yes, our current theoretical result can only be obtained after the training. However, we do think it is of much value. Beyond its theoretical value of the fundamental limit, we are also able to identify the most important factors that impact the pruning performance, i.e., the network flatness (namely, the trace of the Hessian matrix) and the magnitude of weight. If we take into account these factors into the training objective, we think it will offer help for sparsifying the network during training (rather than sparsifying after training).
>
> Furthermore, we do think determining the pruning ratio before the training is a very intersting and important avenue for research.
>
> **Weaknesses 4: The pruning method itself is not novel, it is simply $l_1$ regularized training + magnitude pruning.**
>
> It's true that our pruning algorithm is not novel. Actually, the main purpose of our work is to characterize the fundamental limit of network pruning, thus obtaining insights on the key factors that most impact the pruning.  Our proposed pruning algorithm is a byproduct, which serves as validating our theoretical results.

---

> ### Author Response · Authors · 2023-11-23
> **Response to Reviewer yqv3**
>
> **Question 1: Can the proposed method be used for dense models to predict its pruning rates? If it can be used, could you provide some results?**
>
> We are not quite sure about what "dense models" mean. If it refers to the original network, our proposed theory can definitely predict its pruning limit, since our theory actually makes no assumptions on the model or loss function.
>
> If dense model refer to the one obtained after training, it seems it have been  assumed that after the $l_1$ regularized training, the network gets sparse automatically, which, unfortunatley is not the case. We have responded to this point in Weakness 2. For sake of brevity, we choose to omit here.
>
> **Question 2: Do you try your methods on large-scale datasets like ImageNet? Does the computation of the Hessian Matrix become a bottleneck for large-scale datasets?**
>
> As for the large-scale dataset, due to the limited computational resources in our lab, experiments based on the ImageNet dataset is of difficulty for now. Therefore, we chose to use tiny version of Imagenet for validation, which shows that our theory aligns well with experimental outcomes. Relevant results can be found in Section 5. If this paper was finally accepted and experiments were finished then, we will provide the  validation results on Imagenet  in the final version.
>
> Regarding the computational concern, it's important to note that in the Lanczos-based eigenvalue computation algorithm we employed, direct calculation of the Hessian matrix ${\bf H}$ is actually not required. Instead, the computation only involves ${\bf Hv}$, where ${\bf v}$ is a Rademacher vector,  a rather straightforward operation on the platform of PyTorch. Furthermore, in our experiments, we actually did not utilize the entire dataset, in fact accurate calculation of the theoretical pruning ratios could be achieved with as little as 10\% or even fewer data points.
>
> **Question 3: This is an open question just for discussion. Is it possible for your current theoretical framework to incorporate $\lambda$ to predict the final theoretical pruning rate? Is it possible for your current framework to use the theoretical pruning rate of an early trained model to predict the pruning rate of the fully trained model?**
>
> Thanks for the insightful questions. In fact, we are now striving for developing an algorithm to predict $\lambda$ based on our theoretical results, aiming to establish the correlation between $\lambda$ and the final pruning ratio of the network.
>
> Additionally, the  prospect of using early-stage pruning ratios to predict the final pruning ratio is also an interesting avenue for exploration. Although this aspect is not addressed in the current manuscript, we believe it holds great promise for future research. Furthermore, the prospect of determining the maximum pruning ratio of the network prior to training, for guiding pre-training pruning, is a direction that we're quite interested.

---

### Official Review · Reviewer_VwZk · 2023-11-01

**Soundness:** 3 good
**Presentation:** 3 good
**Contribution:** 3 good
**Rating:** 8
**Confidence:** 2

**Summary:**

This paper delves into the fundamental limit of pruning ratios in deep networks by employing a high-dimensional geometry framework. By adopting this geometric perspective, the paper leverages  tools such as statistical dimension and the Approximate Kinematic Formula to precisely pinpoint the sharp phase transition point of network pruning. The authors bound the maximal portion of weights that can be removed in a network without negatively affecting its performance. They also improve the  spectrum estimation algorithm for very large Hessian matrices when computing the Gaussian width. This work provides some insights into the factors impacting pruning performance and validates findings through experiments. The experiments align well with theoretical results.

**Strengths:**

- The authors investigate the maximum achievable reduction in the number of parameters through network pruning without sacrificing performance, using the perspective of high-dimensional geometry. They study an interesting question and the methodology is inspiring.
- Extensive experiments conducted by the authors demonstrate alignment with their theoretical analyses.
- Their analysis also lends support to commonly observed phenomena, such as the effectiveness of iterative magnitude pruning and the use of regularization.

**Weaknesses:**

The paper is well structured but providing more intuitive descriptions could enhance the article's clarity and help readers follow its logic. Besides, it's not clear whether there are implicit assumptions in the analysis, such as whether it is limited to specific network architectures or loss functions.

The reviewer did not identify significant technical flaws. However, the reviewer is not familiar with the techniques employed in this paper, and the proofs in the supplementary materials have not been thoroughly verified.

**Questions:**

see weaknesses.

---

> ### Author Response · Authors · 2023-11-23
> **Response to Reviewer VwZk**
>
> Thank you very much for the valuable and insightful feedback. Our response are as follows:
>
> **Weaknesses 1: The paper is well structured but providing more intuitive descriptions could enhance the article's clarity and help readers follow its logic.**
>
> Thanks for the suggestion. We have modified the main idea description in a more intuitive manner in the revised version. Specifically, we have provided an intuitive description of the key tool in our work, i.e., the Approximate Kinematics Formula. Please refer to the following two excerpts:
>
> In Section 1:
>
> > Despite the above progress, it however still remains elusive about the *fundamental limit* of network pruning while maintaining the performance. To tackle this problem, we'll take a first principle approach by imposing the sparsity constraint directly on the loss function, thus converting the original pruning limit problem to a set intersection problem, i.e., deciding whether the *$k$-sparse set* intersects with the *loss sublevel set* (i.e., the set of weights whose corresponding loss is no larger than the original loss plus  tolerance $\epsilon$)  and obtaining the smallest value of $k$.
>
> >
> > Intuitively speaking, the larger the loss sublevel set (higher complexity), the smaller the sparse set required for intersection, i.e., the network can be more sparse. To rigorously characterize the complexity of a set, by exploiting the high dimensional nature of DNNs, we are able to harness the notion of *statistical dimension* and *Gaussian width* in high dimensional convex geometry. Through this geometric perspective, it's possible for us to take advantage of the universal *concentration effect*[1]  in the high-dimensional world, so as to get sharp results about the above set intersection problem. In specific, we will exploit the powerful Approximate Kinematics Formula in high-dimensional geometry, which roughly says that for two convex cones, if the sum of their statistical dimension exceeds the ambient dimension, then these two cones would intersect with probability 1, otherwise they would intersect with probability 0. We notice that a sharp phase transition emerge here, thus enabling a precise and succinct characterization of the fundamental limit of network pruning.
>
> and in Section 2:
>
> > To characterize the sufficient and necessary condition of the set (or cone) intersection, the Approximate kinematics Formula is a powerful and sharp result, which basically says that for   two convex cones (or generally, sets), if the sum of their statistical dimension exceeds the ambient dimension, then these two cones would intersect with probability 1, otherwise they would intersect with probability 0.
>
>   [1] Basically, the concentration effect says that a function of a *large amount* of independent (or weakly dependent) variables tends to concentrate to its expectation value. Notable examples include the Johnson-Lindenstrauss lemma and related results in Compressive Sensing.
>
> **Weaknesses 2: Besides, it's not clear whether there are implicit assumptions in the analysis, such as whether it is limited to specific network architectures or loss functions.**
>
> Actually we impose not any assumption about the network models or loss functions within our theoretical framework. In other words, it is highly universal, for example, our result can be easily applied to the Large Language Models (LLMs).

---

### Official Review · Reviewer_QdAJ · 2023-11-01

**Soundness:** 3 good
**Presentation:** 2 fair
**Contribution:** 3 good
**Rating:** 5
**Confidence:** 2

**Summary:**

This paper offers a theoretical exploration of the fundamental limit of network pruning and demonstrates that the derived sparsity level bounds align effectively with empirical observations. They also proposed an improved spectrum estimation algorithm When computing the Gaussian widths of a high-dim and none-convex set. The theoretical analysis provides plenty of insights about previous observations in network pruning, such as the comparison between magnitude and random pruning, one-shot pruning, and the iterative one.

**Strengths:**

- The theoretical analysis presented in the paper aligns seamlessly with experimental results and offers valuable insights for the pruning community.
- The experiments cover a broad spectrum of datasets and model architectures, enhancing the paper's comprehensiveness.
- The notations are clearly defined, and the paper exhibits a logical organization, making it accessible and well-structured.

**Weaknesses:**

- Section 5.3 appears to be somewhat disconnected from the primary focus of the work. Given the widespread use of regularization-based pruning algorithms in the literature (e.g., [1,2]), it might be worth considering how this section better aligns with the core contributions of the paper.

- In Table 3, the numbers in the "Sparsity" column should be subtracted by 100.

- It would enhance the clarity of Figure 3 to employ a log-scale x-axis, which would allow for a more effective visualization of the sharp drop-off region.

[1] https://openreview.net/pdf?id=o966_Is_nPA

[2] https://proceedings.neurips.cc/paper_files/paper/2019/file/4efc9e02abdab6b6166251918570a307-Paper.pdf

**Questions:**

In the era of Large Language Models(LLM), how do the theoretical results align with LLM pruning results[3]?

[3] https://arxiv.org/abs/2306.03805

---

> ### Author Response · Authors · 2023-11-23
> **Response to Reviewer QdAJ**
>
> We sincerely thank you for your insightful feedback. Below are our detailed responses to your concerns:
>
> **Weaknesses 1: Section 5.3 appears to be somewhat disconnected from the primary focus of the work. Given the widespread use of regularization-based pruning algorithms in the literature (e.g., [1,2]), it might be worth considering how this section better aligns with the core contributions of the paper.**
>
> Regularization-based pruning algorithms are indeed not new. For example, in [1] the $l_2$ regularization is employed,  while in [2] the $l_0$-norm is used, however the regularization is not directly imposed on the weight, rather it is on the auxiliary variables.
>
> To the best of our knowledge, our work is the *first* to employ the $l_1$ regularization directly on the weight and *one-shot pruning* is performed, in the spirit of first principles modeling and problem-solving.  Therefore, we think the performance of our proposed $l_1$-regularized one-shot pruning algorithm, which turns out to outperform all typical pruning algorithms, is a valuable complement, to our theoretical result.
>
> [1] https://openreview.net/pdf?id=o966_Is_nPA
>
> [2] https://proceedings.neurips.cc/paper_files/paper/2019/file/4efc9e02abdab6b6166251918570a307-Paper.pdf
>
> **Weaknesses 2: In Table 3, the numbers in the "Sparsity" column should be subtracted by 100.**
>
> In the context of Table 3, "Sparsity" means the pruning ratio, therefore, there is actually no need to subtract it from 100.
>
> **Weaknesses 3: It would enhance the clarity of Figure 3 to employ a log-scale x-axis, which would allow for a more effective visualization of the sharp drop-off region.**
>
> Thanks, we incorporated the suggestions. We have modified the x-axis of Figure 3 to employ non-uniform coordinates, emphasizing the region around the phase transition point.
>
> **Question 1: In the era of Large Language Models(LLM), how do the theoretical results align with LLM pruning results[3]?**
>
> This is an excellent question. In our opinion, our theoretical result aligns well with LLM pruning results in [3], due to the following reasons: 1) Our theory imposes no constraints on network models or loss functions, thus enabling its high universality. 2) The pruning methods in [3] also utilizes one-shot magnitude pruning, similar as ours, therefore its limit (referred as Essential sparsity in [3]) should align with our proposed fundamental limit.
>
> Due to that  currently we lack the required computing resources for running LLMs, we're unable to conduct experiments for validation purpose.
>
> [3] https://arxiv.org/abs/2306.03805

---

### Author Response · Authors · 2023-11-23
**General Reply to All Reviewers**

We sincerely thank all reviewers for the valuable and constructive feedback, which were all incorporated in the revised paper. Here, we summarize the main changes to our paper and all corresponding contents are highlighted in red in the revised paper.

We have adjusted the structure of the manuscript and some definitions to enhance the clarity of the logical flow, without altering the content expressed in the paper.

**Section 1:** We added an overarching explanation for the entire work in a more intuitive manner.

**Section 2:** We moved key notions and results in high dimensional convex geometry to Section 2 and added an intuitive description of the Approximate Kinematics Formula.

**Section 5:** We modified the x-axis of Figure 3 to employ non-uniform coordinates, emphasizing the region of sharp drop-off.

---

### Meta-Review · Area_Chair_ksyA · 2023-12-15

**Metareview:**

While the initial reviews show some luke-warm to moderate support for the work, the main result on the dependence on square of Gaussian width is a widely known result (see ref below), numerous papers have been written on the theme, and the work does not even cite the most influential papers on the them. Given that, the novelty and new contributions of the work is limited, the authors seem unaware of highly impactful papers and several books on the theme.

Primary ref, which has ~1500 citations, many of which one way or other works with the squared Gaussian width and related results https://arxiv.org/abs/1012.0621

More broadly, the books by (1) Talagrand, (2) Vershynin, and (3) Wainwright have extensive coverage on related results, including width of ellipsoids (e.g., see Talagrand's book), extensions of Gordon's results, etc. None of this is cited. Some of the computational aspects of the work are arguably interesting, but the main technical results are not novel.

**Justification For Why Not Higher Score:**

The work misses significant existing work on the topic, which covers the main technical result presented in the paper.

**Justification For Why Not Lower Score:**

n/a

---

### Decision · Program_Chairs · 2024-01-16

Reject